# Carbon nano-onion-mediated dual targeting of P-selectin and P-glycoprotein to overcome cancer drug resistance

Hai Wang [1,2,3,7✉], Yutong Liang[1,7], Yue Yin[2,7], Jie Zhang[2], Wen Su[2], Alisa M. White[1], Bin Jiang[1], Jiangsheng Xu[1], Yuntian Zhang[1], Samantha Stewart[1], Xiongbin Lu [4] & Xiaoming He [1,5,6✉]

The transmembrane P-glycoprotein (P-gp) pumps that efflux drugs are a major mechanism of cancer drug resistance. They are also important in protecting normal tissue cells from poisonous xenobiotics and endogenous metabolites. Here, we report a fucoidan-decorated silica-carbon nano-onion (FSCNO) hybrid nanoparticle that targets tumor vasculature to specifically release P-gp inhibitor and anticancer drug into tumor cells. The tumor vasculature targeting capability of the nanoparticle is demonstrated using multiple models. Moreover, we reveal the superior light absorption property of nano-onion in the near infrared region (NIR), which enables triggered drug release from the nanoparticle at a low NIR power. The released inhibitor selectively binds to P-gp pumps and disables their function, which improves the bioavailability of anticancer drug inside the cells. Furthermore, free P-gp inhibitor significantly increases the systemic toxicity of a chemotherapy drug, which be resolved by delivering them with FSCNO nanoparticles in combination with a short low-power NIR laser irradiation.

[1] Fischell Department of Bioengineering, University of Maryland, College Park, MD 20742, USA. [2] CAS Key Laboratory for Biomedical Effects of Nanomaterials & Nanosafety, CAS Center for Excellence in Nanoscience, National Center for Nanoscience and Technology, 100190 Beijing, China. [3] University of Chinese Academy of Sciences, 100049 Beijing, China. [4] Department of Medical and Molecular Genetics and Melvin and Bren Simon Cancer Center, Indiana University School of Medicine, Indianapolis, IN 46202, USA. [5] Marlene and Stewart Greenebaum Comprehensive Cancer Center, University of Maryland, Baltimore, MD 21201, USA. [6] Robert E. Fischell Institute for Biomedical Devices, University of Maryland, College Park, MD 20742, USA. [7] These authors contributed equally: Hai Wang, Yutong Liang, Yue Yin. ✉email: wanghai@nanoctr.cn; shawnhe@umd.edu

Resistance to chemotherapy or molecularly targeted therapies greatly impedes the efficacy of cancer treatments[1,2]. The drug-resistant cells can survive conventional cancer therapies and may cause tumor recurrence and/or development of distant metastases, which are the major reasons for cancer-related death[3,4]. Consequently, multidrug-resistant cancer cells have attracted a great deal of attention in the field of oncology for several decades[5]. Many different mechanisms may enable or promote multidrug resistance, but drug efflux pumps have been considered as one of the key mechanisms, which are located on the cell membrane to efflux anticancer drugs from cancer cells[6,7]. As a result, various kinds of nanoparticles have been developed to deliver anticancer drugs into multidrug-resistant cancer cells because the drug efflux pumps cannot efflux nanoparticles from cancer cells[8,9]. Unfortunately, studies have shown that drugs can still be easily pumped out of cells once they are released from nanoparticles, which limits the retention/bioavailability of drugs inside multidrug-resistant cancer cells for effective cancer destruction[10,11]. Thus, direct inhibition of the drug efflux pumps may be an effective strategy for overcoming the multidrug resistance of cancer.

An important advancement in the understanding of the drug efflux pumps is the identification of the transmembrane P-glycoprotein (P-gp) in the plasma membrane of multidrug-resistant cancer cells[12,13]. The P-gp multidrug transporter is an ATP-binding cassette (ABC) protein that has been implicated in multiple types of multidrug-resistant cancers[7,14]. Thus, inhibiting the function of P-gp could be an effective approach to improve the retention/bioavailability of therapeutic agents inside cancer cells. This can be done by using inhibitor to block the P-gp efflux. Indeed, coadministration of the therapeutic agents and P-gp inhibitors in their free forms may reduce the expulsion of the agents by P-gp and increase the intracellular drug concentration[15,16]. Nevertheless, P-gp pumps also play a crucial physiological role in protecting tissues from poisonous xenobiotics and endogenous metabolites that are widely distributed throughout the body[17–19]. For example, the P-gp pumps are a major mechanism of the blood-brain barrier (BBB) and intestinal wall to minimize the entry of any toxic substances into the brain and blood, respectively (Fig. 1a)[20,21]. In other words, cellular processes in the body are benefiting from the protection of P-gp and nontargeted systemic administration of P-gp inhibitor in its free form may cause unexpected side effects[22–24].

To tackle these challenges, here we report a tumor vasculature targeted nanovesicle to codeliver P-gp inhibitor and chemotherapeutic drug. This nanoplatform (FSCNO) is made of carbon nano-onion (CNO) and silica (S) surface-decorated with fucoidan (F), which can specifically bind to P-selectin that is overexpressed on tumor vasculature (Fig. 1b, right)[25,26]. As a result of this active tumor vasculature targeting and their passive tumor-targeting capacity due to the enhanced permeability and retention (EPR) effects of tumor vasculature compared to normal vasculature[27], the FSCON nanoparticles are anticipated to preferentially accumulate in tumor rather than normal tissues after systemic administration in vivo (Fig. 1b, right). Moreover, the release of P-gp inhibitor and chemotherapeutic drug from FSCON nanoparticles is negligible in normal tissue in the absence of an external stimulus (low-power laser irradiation). By minimizing the accumulation of FSCON nanoparticles in normal organs and keeping the inhibitors inside the nanoparticles, the function of P-gp efflux in normal organs to exclude poisonous xenobiotics and endogenous metabolites is minimally affected (Fig. 1b, c, left). In contrast, the release of P-gp inhibitor and chemotherapeutic drugs can be triggered and precisely controlled with low-power near infrared (NIR, 800 nm) laser irradiation in tumor to overcome the multidrug resistance of tumor cells (Fig. 1b, c, right).

## Results

**Preparation and characterization of nanoparticles.** Carbon nano-onions (CNOs), also known as multilayer fullerenes, have been reported for cancer cell imaging[28,29]. Unfortunately, due to their highly hydrophobic nature, CNOs have poor solubility in water or even commonly used organic solvents (e.g., benzene and toluene) and they tend to form large aggregates in these solvents (Supplementary Fig. 1a). This greatly limits the biomedical applications of CNOs. To address this issue, CNOs were treated with tetraethyl orthosilicate (TEOS) to form silica-CNO (SCNO) nanoparticles (Fig. 2a) with increased solubility and drug loading capacity. In order to decorate them with fucoidan for targeting tumor vasculature, the SCNO nanoparticles were further reacted with (3-Aminopropyl) trimethoxysilane (APTMS) to produce the ASCNO intermediates. Fucoidan was then coated on the surface of the ASCNO intermediates through its electrostatic interaction with APTMS to form the resultant FSCNO nanoparticles (Fig. 2a). Transmission electron microscopy (TEM) images of CNOs are given in Fig. 2b. The nanoparticles readily form aggregates and the nano-onion structure (~50 nm in diameter) can be observed only at the edge of the aggregates at higher magnification (right, Fig. 2b). The onion-like structure is observable in the high-resolution TEM (HRTEM) images (Supplementary Fig. 2). Interestingly, the hybrid SCNO nanoparticles are smaller than the CNOs and their nanostructure can be clearly observed as shown in Supplementary Fig. 3. This is probably due to the reaction between TEOS and CNO according to the Fourier-transform infrared spectroscopy (FTIR) spectrum (i.e., the formation of the Si–O–C bond, Supplementary Fig. 4a), making the CNO structure more compact. After coating the SCNO nanoparticles with fucoidan, the FSCNO nanoparticles of ~20 nm with clear nanostructure and little aggregation can be obtained (Fig. 2c). The successful decoration of fucoidan is confirmed with the surface zeta potential data of the SCNO, ASCNO, and FSCNO nanoparticles. As shown in Supplementary Fig. 5, the zeta potential is negative for the SCNO nanoparticles at room temperature ($-23.7 \pm 1.8$ mV) but becomes positive ($38.5 \pm 2.4$ mV) after modifying them with positively charged APTMS to form the ASCNO intermediates. It returns to negative ($-38.0 \pm 4.1$ mV) after coating the ASCNO nanoparticles with the negatively charged fucoidan, suggesting fucoidan is successfully decorated on the FSCNO nanoparticles. Importantly, unlike the CNOs, the resultant FSCNO nanoparticles have high solubility and stability in both deionized water and cell culture medium (Supplementary Fig. 1a). This is further confirmed with the Tyndall effect that is a result of light scattering by nanoparticles, by shining a red laser beam through various solutions of the CNOs and FSCNO nanoparticles. As shown in Supplementary Fig. 1b, a discernible light track can be observed only in the solutions of FSCNO nanoparticles (in deionized water and cell culture medium).

To inhibit the pumping capability of P-gp, a third-generation inhibitor (HM30181A, HM in short) was used in this study that can selectively and potently inhibit the P-gp function[30]. However, HM is insoluble in water and was used only through oral delivery in previous studies to inhibit P-gp in the intestinal endothelium. It has never been used to inhibit the P-gp in multidrug-resistant cancer cells. Both the hydrophobic HM and hydrophilic doxorubicin hydrochloride (DOX, a widely used chemotherapy drug) were encapsulated in the FSCNO nanoparticles to form FSCNO-DH nanoparticles, with an encapsulation efficiency of $68.5 \pm 1.7\%$ for HM and $89.0 \pm 1.5\%$ for DOX at a feeding ratio of 1:20 (HM or DOX: nanoparticles). The corresponding loading content of HM and DOX is $3.3 \pm 0.1\%$ and $4.3 \pm 0.1\%$, respectively. To confirm the FSCNO nanoparticles have spaces for encapsulating drugs, nitrogen gas ($N_2$) sorption

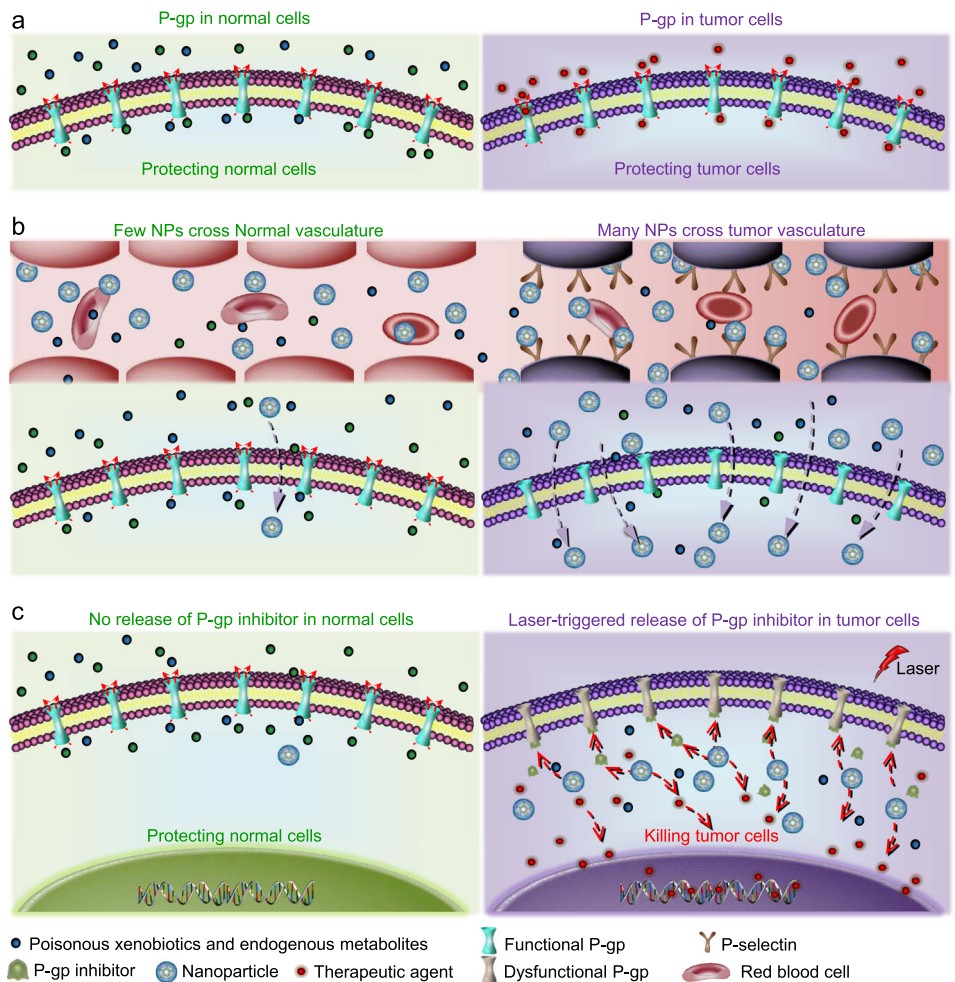

**Fig. 1 A schematic illustration of the strategy for targeting tumor vasculature and inhibiting the efflux pumps in drug-resistant tumor. a** The P-glycoprotein (P-gp) efflux pumps are beneficial for normal cells to keep poisonous xenobiotics and endogenous metabolites outside of them. Unfortunately, drug-resistant cancer cells also use P-gp to exclude therapeutic agents. Therefore, nontargeted systemic administration of P-gp inhibitor may cause unexpected side effects. **b** In this study, a tumor vasculature targeted nanoparticle (NP) to codeliver P-gp inhibitor and chemotherapeutic drug is developed to specifically inhibit the P-gp function in tumor. The nanoparticle can preferentially accumulate in tumors as a result of both the P-selectin mediated active targeting and the enhanced permeability and retention (EPR) effect mediated passive targeting. **c** With near infrared (NIR, 800 nm) laser irradiation, the nanoparticles could release the therapeutic agents and P-gp inhibitor in tumor cells. P-gp inhibitor can then competitively bind to P-gp pumps and disable their function. Importantly, the P-gp pumps in normal cells should be minimally affected due to the little accumulation of the nanoparticles in them and the little release of the P-gp inhibitor from the nanoparticles even if a small number of nanoparticles might enter normal tissue or cells.

measurements were carried out to investigate both the surface area and pore size distribution within the nanoparticles. The $N_2$ adsorption isotherm of the FSCNO nanoparticles (Supplementary Fig. 6a) is typical of the type IV isotherm that indicates the micro- or mesoporous feature according to the International Union of Pure and Applied Chemistry (IUPAC) classification[31]. The surface areas within the FSCNO nanoparticles are 250.3 m² g⁻¹ and the average pore size is 3.2 nm with two peaks at ~1.4 nm and ~3.1 nm (Supplementary Fig. 6b). This porous structure within the FSCNO nanoparticles renders their capability of drug loading. The loading of drugs inside the nanoparticles rather than on their external surface is further supported by the zeta potential data. As shown in Supplementary Fig. 7a, the zeta potential of the two drugs-laden FSCNO-DH nanoparticles is similar to that of the FSCNO nanoparticles with no drug (Supplementary Fig. 5c), suggesting the drugs are not coated on the surface of the nanoparticles.

The FSCNO-DH nanoparticles is ~28 nm (in diameter) according to the dynamic light scattering (DLS) measurements (Fig. 2d). The successful encapsulation of HM and DOX was

further confirmed with UV-Vis absorbance spectrometry. As shown in Fig. 2e, DOX has a strong peak at ~480 nm whereas HM30181A has strong absorbance between 350 and 400 nm. Compared with FSCNO nanoparticles, the DOX-laden nanoparticles (FSCNO-D) have a strong peak at ~480 nm due to DOX in the nanoparticles (Fig. 2f). Similarly, compared to the FSCNO-D nanoparticles, the absorbance between 350 and 400 nm is elevated for the FSCNO-DH nanoparticles because of the strong absorbance of HM30181A in this region (Fig. 2f). The fluorescence spectra show that FSCNO-D nanoparticles and free DOX have the same fluorescence emission peak at ~590 nm after excitation with a 480 nm laser (Fig. 2g), although the fluorescence intensity is decreased compared with free DOX. The latter is probably because of self-quenching of fluorescence and π–π stacking between FSCNO nanoparticles and DOX in the FSCNO-DH nanoparticles[32]. HM does not have fluorescence and has no effect on the fluorescence of DOX in FSCNO-DH nanoparticles (Fig. 2g). Collectively, these data show that both P-gp inhibitor and chemotherapeutic drug are successfully encapsulated in the FSCNO-DH nanoparticles.

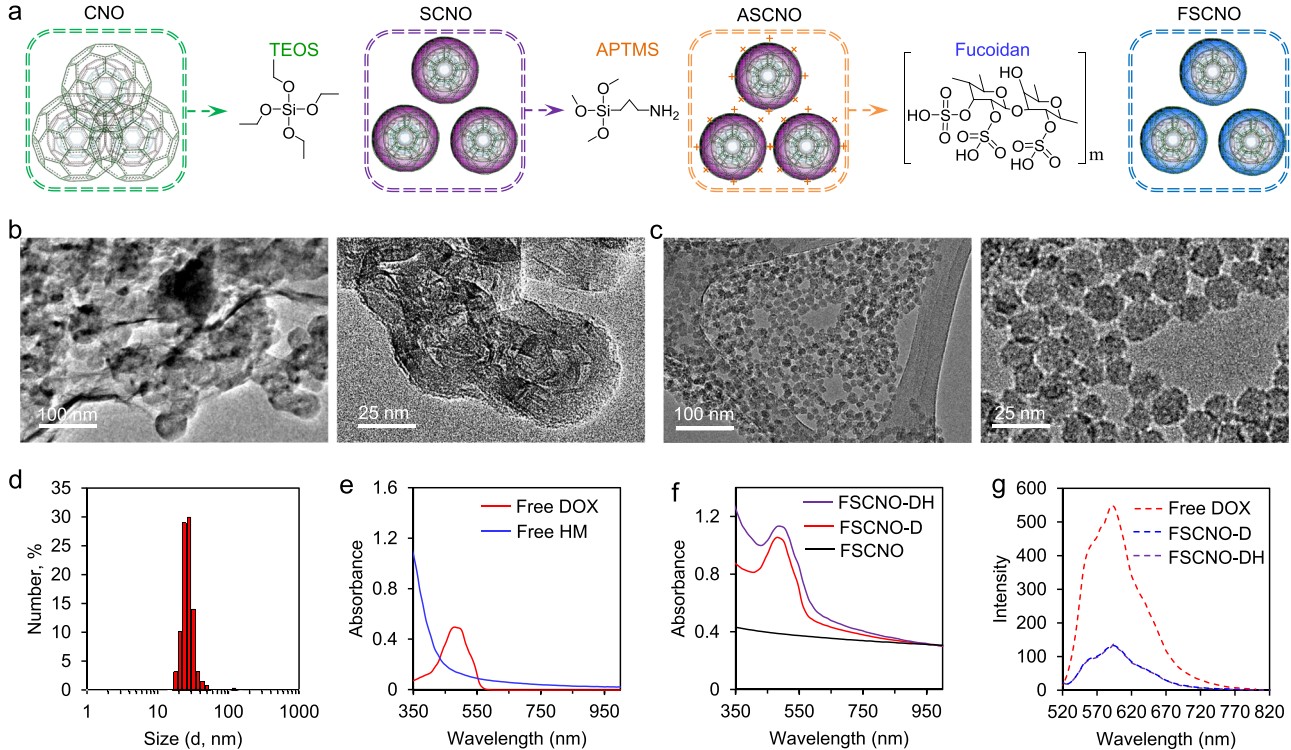

**Fig. 2 Synthesis and characterization of nanoparticles. a** A schematic illustration of the procedure for preparing the silica-carbon nano-onion (SCNO) nanoparticles using carbon nano-onions (CNOs) and tetraethyl orthosilicate (TEOS), modifying the SCNO nanoparticles with (3-Aminopropyl) trimethoxysilane (APTMS) to form ASCNO nanoparticles, and coating the ASCNO nanoparticles with fucoidan to produce the FSCNO nanoparticles. **b, c** Transmission electron microscopy (TEM) images of the CNOs (**b**) and FSCNO nanoparticles (**c**). **d** Size (in diameter) distribution of the FSCNO nanoparticles determined by dynamic light scattering (DLS) at room temperature. **e** UV-Vis absorbance of free doxorubicin hydrochloride (DOX) in deionized water and HM30181A (HM) in DMSO solution showing DOX has an absorbance peak at 485 nm while HM has strong absorption around 350 nm. **f** UV-Vis absorbance of FSCNO-D nanoparticles in deionized water showing an absorbance peak of DOX and FSCNO-DH in deionized water nanoparticles showing high absorbance at ~350 nm, suggesting the successful encapsulation of DOX and HM. **g** Fluorescence spectrophotometry of FSCNO-D and FSCNO-DH nanoparticles together with free DOX showing successful encapsulation of DOX in the two nanoparticles with a fluorescence peak at ~590 nm.

**Photothermal effect of FSCNO nanoparticles**. Due to its capability of deep tissue penetration, NIR laser has been used as an external stimulus to trigger the release of drugs encapsulated in gold and carbon nanomaterials with strong absorption in the NIR region[33]. Therefore, we investigated the UV-Vis absorbance spectrum of FSCNO nanoparticles at different concentrations. Interestingly, the absorbance of FSCNO nanoparticles decreases only slightly from ultraviolet to NIR region (Fig. 3a). We also synthesized FSCNT and FSGO nanoparticles with carbon nanotube (CNT) and graphene oxide (GO) that have been widely studied for photothermal applications, for which the same procedure given in Fig. 2a were used except that CNOs was replaced with CNTs and GOs, respectively. As shown in Supplementary Fig. 8, scanning electron microscopy (SEM) image of CNTs shows their tube structure is changed after reacted with silica and coated with fucoidan to form the FSCNT nanoparticles. Although the sheet structure of GO is difficult to identify in the SEM image because they are very thin, the thickness of the sheet structure is significantly increased and clearly visible after reacted with silica and coated with fucoidan to form the FSGO nanoparticles (Supplementary Fig. 8). The FTIR spectra of the SCNT and SGO exhibit absorption bands corresponding to Si–O–Si ($\nu_s$ at 800 and $\nu_{as}$ at 1080–1200 cm$^{-1}$, where the subscripts s and as represent symmetric and asymmetric, respectively) and Si–O–C ($\nu_s$ at 954 and $\nu_{as}$ at 1070 cm$^{-1}$), suggesting the formation of silica structure and the reaction between TEOS with CNT or GO (Supplementary Fig. 4b–c). The absorbance of FSCNT and FSGO nanoparticles

decreases significantly from the ultraviolet to NIR region for all the concentrations studied (Fig. 3a). In particular, the absorbance of the FSCNO nanoparticles in the NIR region is 2–3 times higher than that of the FSCNT and FSGO nanoparticles when the concentration is more than 0.25 mg ml$^{-1}$, implicating the superior photothermal property of the FSCNO nanoparticles. Indeed, the temperature of FSCNO nanoparticle solution increases faster than that of the FSCNT or FSGO nanoparticle solution at the same concentrations under NIR laser irradiation (1 W cm$^{-2}$, Fig. 3b). The photothermal effects of the FSCNO nanoparticles is further demonstrated in centrifuge tube with near infrared thermography (Supplementary Fig. 9). Interestingly, the photothermal effect of the FSCNO nanoparticles can also be observed in their dry state. For this study, the FSCNO nanoparticle solutions at various concentrations (1.8–280 ng mm$^{-2}$) was dropped on paper, completely dried, and irradiated with NIR laser (1 W cm$^{-2}$). As shown in Fig. 3c, the paper was observed to ignite in ~1–3 s in the area with 7 ng mm$^{-2}$ or more FSCNO nanoparticles, while burning was seen at ~10 s in the area with 3.5 ng mm$^{-2}$ nanoparticles and it was not seen in the area with 1.8 ng mm$^{-2}$ nanoparticles in 20 s. The stability of the photothermal effects was examined with three cycles of NIR irradiation (1 W cm$^{-2}$, on-off: 2–10 min) of the FSCNO nanoparticle solutions continuously. As shown in Fig. 3d, the temperature change in the solutions is similar for all the three cycles. This is not surprising as the laser does not affect the integrity of the nanoparticles (Supplementary Fig. 10). Importantly, the release of P-gp

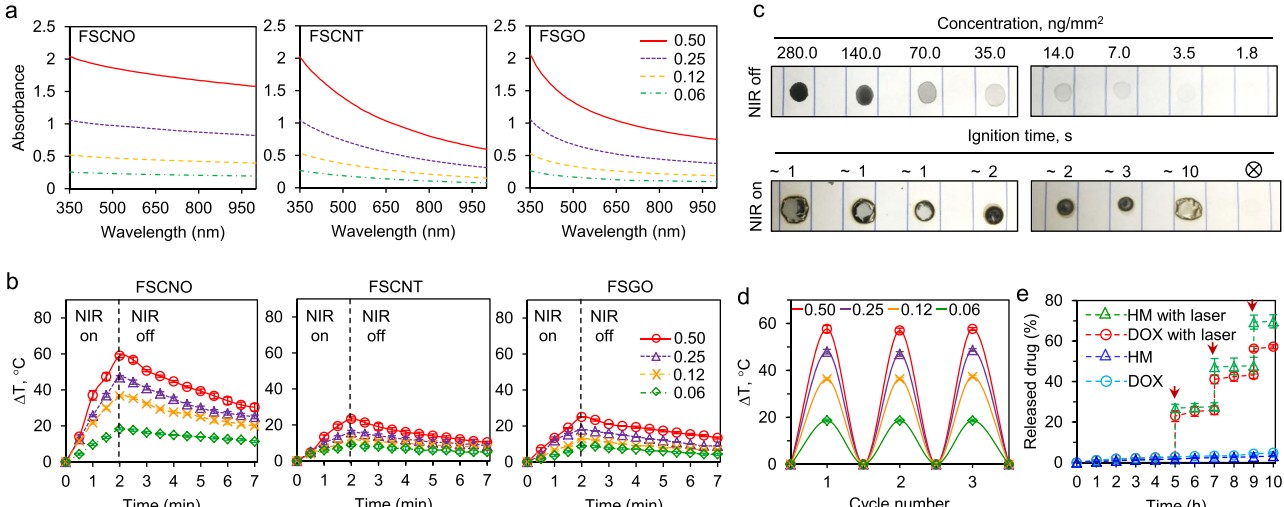

**Fig. 3 Photothermal effect of FSCNO nanoparticles and controlled release of therapeutic agents. a** UV-Vis absorbance of FSCNO, FSCNT, and FSGO nanoparticles in water at different concentrations (0.06, 0.12, 0.25, and 0.50 mg ml$^{-1}$) showing the FSCNO nanoparticles have higher absorption in the NIR region than the FSCNT and FSGO nanoparticles. **b** The temperature in aqueous solutions of FSCNO nanoparticles increases faster than that in the aqueous solutions of FSCNT and FSGO nanoparticles upon NIR irradiation (1.0 W cm$^{-2}$) for 2 min at different concentrations (0.06, 0.12, 0.25, and 0.5 mg ml$^{-1}$). **c** Superior photothermal effect of FSCNO nanoparticles. The FSCNO nanoparticles solution (15 µl) was dropped on papers and completely dried with concentrations from 1.8 to 280 ng mm$^{-2}$. Then, the nanoparticles were irradiated with NIR laser (1 W cm$^{-2}$) and the time of ignition was recorded. Paper burning was observed in 1–10 min for all the concentrations except the lowest concentration (1.8 ng mm$^{-2}$). **d** Stability of the FSCNO nanoparticles in aqueous solutions during three cycles of NIR laser irradiation. The FSCNO nanoparticles (0.06, 0.12, 0.25, and 0.50 mg ml$^{-1}$) were irradiated for 2 min and then passively cooled down for 10 min in each cycle. **e** The release of DOX and HM from the FSCNO nanoparticles is slow but a quick release can be precisely triggered and controlled with near infrared (NIR) laser irradiation (0.5 W cm$^{-2}$) for 1 min. The arrows indicate the laser irradiation treatment at three different time points. Error bars represent ± s.d. ($n = 3$ independent runs).

inhibitor and chemotherapeutic drug from the nanoparticles is minimal at least for 10 h and can be precisely controlled with NIR irradiation (Fig. 3e), to minimize their uncontrolled release in normal tissue.

**P-gp inhibition and anticancer capacity of FSCNO-DH nanoparticles in vitro.** Two types of multidrug-resistant human cancer (NCI/ADR-RES and A2780ADR) cells were used in this study together with a nondrug resistant type of human cancer (OVCAR-8) cells for control. We first confirmed the expression of P-gp in NCI/ADR-RES and A2780ADR cells but not the OVCAR-8 cells with confocal microscopy. As shown in Supplementary Fig. 11, immunostaining of the P-gp can be clearly observed in the cell membrane of both NCI/ADR-RES and A2780ADR cells but not the OVCAR-8 cells. Presumably due to the efflux function of the P-gp pumps, the fluorescence of DOX is not observable in both NCI/ADR-RES and A2078ADR cells after incubated them with free DOX for 6 h. In contrast, free DOX (10 µg ml$^{-1}$) can be observed predominantly in the cell nuclei when it is co-incubated with various concentrations of free HM (1 and 5 µg ml$^{-1}$ for Supplementary Fig. 12 and 10 µg ml$^{-1}$ for Fig. 4a), and the fluorescence in nuclei is stronger for cells treated with 5 or 10 µg ml$^{-1}$ HM than 1 µg ml$^{-1}$ HM. These data indicate that the HM indeed can inhibit the function of P-gp pumps in a dose-dependent manner. Next, we treated NCI/ADR-RES and A2078ADR cells with DOX-laden FSCNO (FSCNO-D) nanoparticles and irradiated with laser (FSCNO + L, 0.5 W cm$^{-2}$). Although the overall DOX fluorescence is increased after laser irradiation, the DOX fluorescence in nuclei is still weak (Fig. 4b). This is presumably because the P-gp could still pump out the released drugs. In contrast, the DOX fluorescence in nuclei is highly increased after NIR irradiation of cells treated with the FSCNO-DH nanoparticles (Fig. 4c). As a control, free DOX can enter the OVCAR-8 cells regardless of treatment with or without

HM (Supplementary Fig. 13) and there is no obvious difference in the distribution of DOX in the OVCAR-8 cells treated with FSCNO-D and FSCNO-DH (Supplementary Fig. 13), suggesting the HM has minimal impact on the drug uptake of the nondrug resistant cancer cells. These results are further confirmed with flow cytometry. As shown in Supplementary Fig. 14a–b, the DOX in NCI/ADR-RES and A2780ADR cells is minimal if they were treated with free DOX and greatly increased when free DOX is combined with HM (DOX+HM) for treating the cells. The DOX fluorescence intensity is stronger in FSCNO-DH+L than FSCNO-DH treated cells, probably due to the self-quenching effect of DOX fluorescence in the FSCNO nanoparticles (Fig. 2g). Similar to the microscopy data, HM has minimal effect on the DOX uptake of OVCAR-8 cells (Supplementary Fig. 14c).

The in vitro anticancer capacity of FSCNO-DH nanoparticles was conducted with NCI/ADR-RES, A2780ADR, and OVCAR-8 cells. Compared with OVCAR-8 cells, free DOX is inefficient to kill NCI/ADR-RES and A2780ADR cells (Fig. 4d), due to the drug-resistant nature of the NCI/ADR-RES and A2780ADR cells. The efficacy of DOX in killing NCI/ADR-RES and A2780ADR cells is significantly enhanced when coadministrated with 1–10 µg ml$^{-1}$ HM (dissolved with DMSO at 1 mg ml$^{-1}$ in stock). Of note, the efficacy of DOX in killing OVCAR-8 cells also increased when coadministrated with HM (5 and 10 µg ml$^{-1}$), which is probably due to the DMSO used to dissolve HM. As shown in Supplementary Fig. 15, DMSO at the concentration of 5 or 10 µl ml$^{-1}$ could kill some of all the three types of cells. Overall, HM of 5 ug ml$^{-1}$ is enough to block the P-gp function and used for the following studies. Although using FSCNO-D nanoparticles can enhance the anticancer effect as a result of improved delivery of DOX into the multidrug-resistant cells, the enhancement is limited for the two drug-resistant cells (Fig. 4d). To check the effect of NIR irradiation on the in vitro anticancer capacity of the nanoparticles, the laser powers at 0.05, 0.2, 0.5, and 1 W cm$^{-2}$ were studied. The results indicate that only the highest power

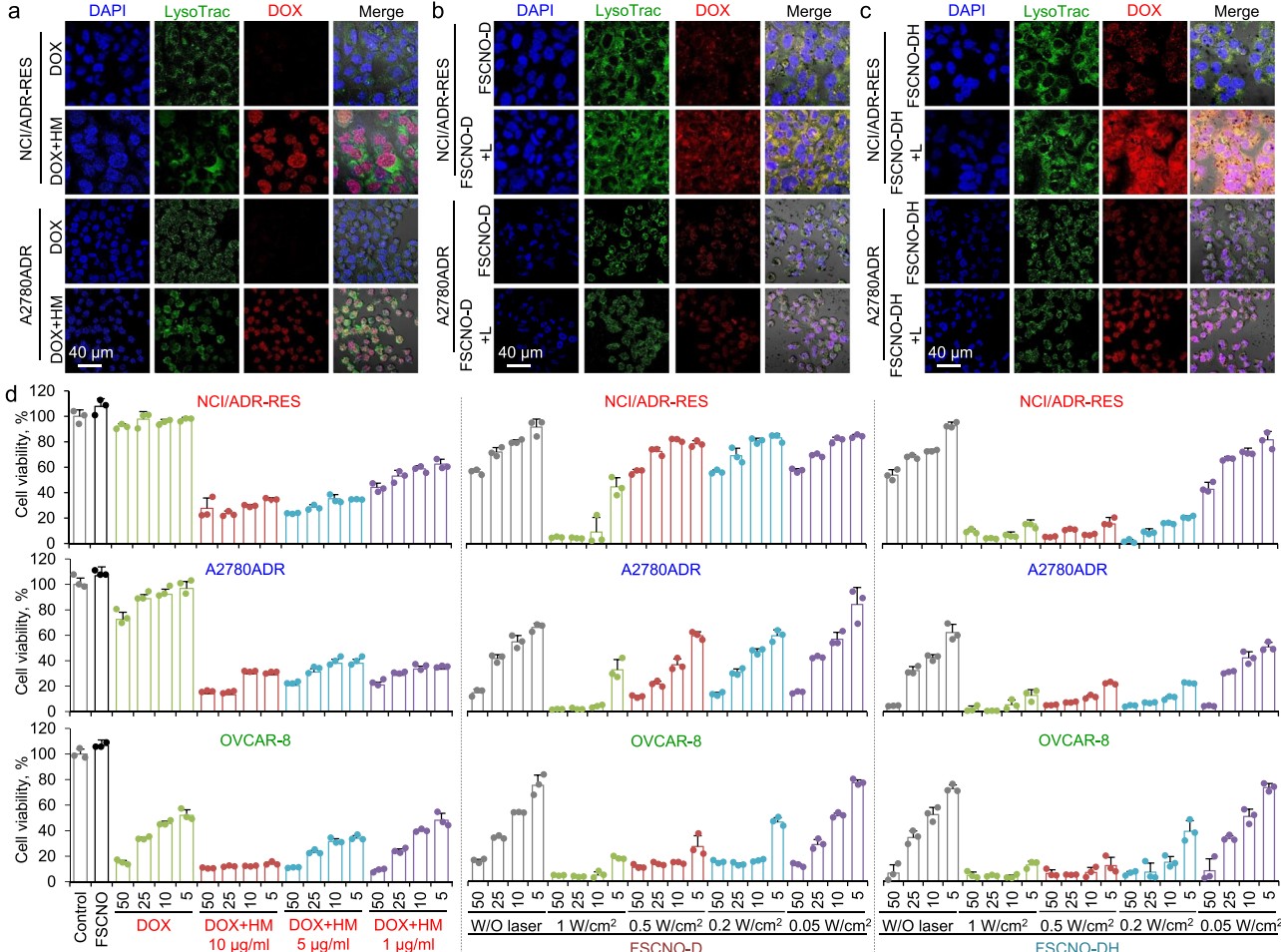

**Fig. 4 Capacity of the FSCNO-DH nanoparticles in overcoming cancer drug resistance and killing drug-resistant cancer cells in vitro. a** Confocal images of DOX in NCI/ADR-RES and A2780ADR multidrug-resistant cancer cells treated with free DOX or combined with HM (DOX+HM, 10 μg ml$^{-1}$ for both DOX and HM). The data show that DOX can enter the multidrug-resistant cancer cells when combined with P-gp inhibitor (HM). LysoTrac: LysoTracker Green. **b, c** Confocal images of NCI/ADR-RES and A2780ADR multidrug-resistant cancer cells treated with FSCNO-D (**b**) or FSCNO-DH (**c**) nanoparticles without or with NIR laser (L) irradiation (FSCNO-D+L or FSCNO-DH+L, 0.5 W cm$^{-2}$ for 1 min). The data show that most of the DOX are distributed in the cytosol for cells treated with FSCNO-D with or without laser irradiation and FSCNO-DH nanoparticles in the absence of NIR laser irradiation while DOX overlaps with DAPI in the cell nuclei for cells treated with FSCNO-DH nanoparticles after irradiated with laser. **d** NCI/ADR-RES, A2780ADR, and OVCAR-8 cancer cells treated with free DOX; free DOX combined with HM at concentrations of 1, 5, and 10 μg ml$^{-1}$; FSCNO-D or FSCNO-DH nanoparticles without or with NIR laser irradiation of different powers for 2 min. The data show that HM is crucial to enhance the anticancer effect of DOX in either free DOX or DOX-laden FSCNO nanoparticles. Error bars represent ± s.d. ($n = 3$ independent runs).

(1 W cm$^{-2}$) combined with FSCNO-D nanoparticles can effectively kill all the three different types of cancer cells. However, this is not because of the anticancer effect of DOX as the empty FSCNO nanoparticles could effectively kill all the three different types of cells under the laser irradiation at 1 W cm$^{-2}$ (Supplementary Fig. 16a). Since the NIR irradiation alone does not compromise the viability of cells at the different laser powers (Supplementary Fig. 16b), the observed anticancer effect is probably due to the aforementioned superior photothermal effect of the FSCNO nanoparticles (Fig. 3 and Supplementary Fig. 9). In contrast, FSCNO-DH nanoparticles (HM: 5 μg ml$^{-1}$) could efficiently destroy all the three types of cells at a laser power as low as 0.2 W cm$^{-2}$ (Fig. 4d). This is probably due to the superior photothermal effect of FSCNO nanoparticles, which could effectively trigger the release of HM to inhibit the P-gp and enhance the retention of DOX in the drug-resistant cells for the observed effective destruction of the cells. Therefore, the laser power of 0.2 W cm$^{-2}$ is used for the following studies and the photothermal effect of the FSCNO nanoparticles at various

concentrations (0.06–0.50 mg ml$^{-1}$) under this laser power is given in Supplementary Fig. 17.

**Tumor vasculature targeting**. To deliver P-gp inhibitor and DOX into tumor rather than normal tissue, the FSCNO nanoparticles were designed to target the tumor vasculature through P-selectin as aforementioned (Fig. 1b). To confirm this, we activated the human umbilical vein endothelial cells (HUVECs) with human tumor necrosis factor-alpha (TNF-α) to mimic the tumor vasculature[34]. As shown in Fig. 5a, the expression of P-selectin is evident in activated HUVECs (aHUVECs) while it is negligible in the HUVECs without activation. The targeting capability of SCNO-DH or FSCNO-DH nanoparticles was then investigated by incubating them with 2D-cultured aHUVECs for 3 h at 4 °C. Indeed, the FSCNO-DH nanoparticles could efficiently bind with P-selectin on the aHUVECs while the binding between SCNO-DH nanoparticles and aHUVECs is not observable (Fig. 5b), showing the crucial role of the fucoidan on the surface of the

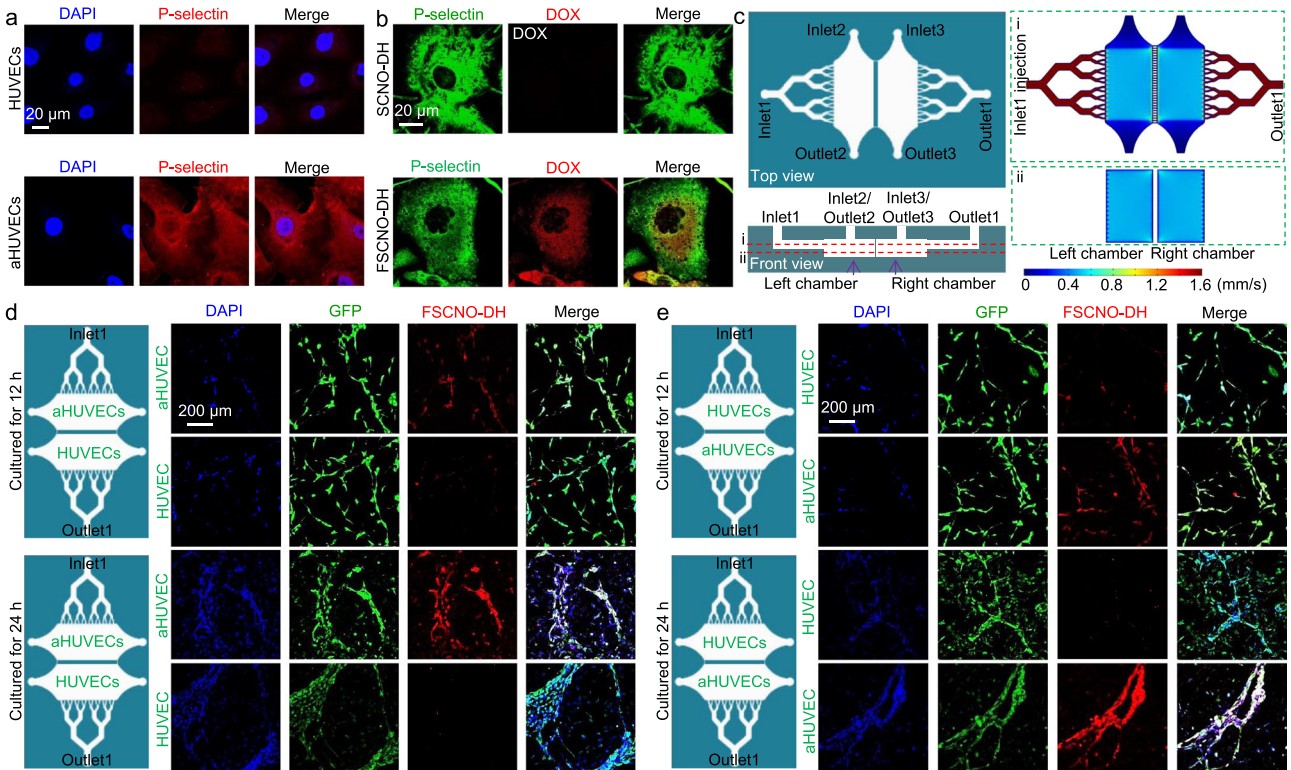

**Fig. 5 P-selectin targeting capability of the FSCNO nanoparticles under both static and dynamic/microfluidic cultures. a** Confocal images showing the expression of P-selectin in activated human umbilical vein endothelial cells (aHUVECs) but not in HUVECs without activation. **b** Confocal images of precooled aHUVECs after incubated with SCNO-DH and FSCNO-DH nanoparticles (10 µg ml$^{-1}$ for DOX and 5 µg ml$^{-1}$ for HM). The cells were precooled in ice for 3 h to stop its metabolic and uptake activity. **c** Top and side views of the microfluidic device for studying the P-selectin targeting capability of the FSCNO-DH nanoparticles under dynamic culture condition. Detached cells can be injected into device through inlet2 or inlet3 and cultured in the chambers. Medium containing nanoparticles can be injected through inlet1 and flows out the device via outlet1. Also shown are the computational modeling results of velocity distribution at the middle planes of the channels and chambers, in the top and bottom PDMS parts as indicated by the red dashed lines i and ii, respectively. The data show that the velocity of injected nanoparticle solution (inlet1) is nearly homogeneous in the two chambers. **d, e** Scheme and confocal images of green fluorescence protein (GFP)-expression HUVECs (in the chamber next to outlet1 for **d** and in the chamber next to inlet1 for **e**) and aHUVECs (in the chamber next to inlet1 for **d** and in the chamber next to outlet1 for **e**) cultured in the microfluidic device for 12 and 24 h after binding with FSCNO-DH nanoparticles in the perfusion medium for 3 h. The data indicate that the FSCNO-DH nanoparticles can target the aHUVECs efficiently regardless of the culture conditions (i.e., static versus dynamic, and for the latter: close or away from the inlet1).

FSCNO-DH nanoparticles in targeting the aHUVECs. The latter is probably via the binding between fucoidan and P-selectin overexpressed on the aHUVECs. This is further supported by the data showing less P-selectin antibody could bind with the FSCNOs nanoparticle-treated aHUVECs than the SCNO-DH nanoparticle-treated aHUVECs (Fig. 5b), probably because some of the P-selectins have been occupied by the FSCNO-DH nanoparticles and hard for the antibody to access.

To further confirm the targeting capability of FSCNO nanoparticles, we fabricated a polydimethylsiloxane (PDMS) microfluidic device for culturing the aHUVECs and HUVECs under continuous dynamic perfusion (Fig. 5c and Supplementary Fig. 18a) to mimic the in vivo blood perfusion in blood vessels. The HUVECs and aHUVECs were cultured in two different and parallel chambers in this device with independent inlets and outlets (inlet2–outlet2 or inlet3–outlet3) for sample injection and the two chambers are separated with a fluid-passable partition wall (Fig. 5c and Supplementary Fig. 18b). Computational modeling data show that fluid flow is almost independent in each chamber during cell injection from inlets 2 or 3 (Supplementary Fig. 18c), making this device ideally suitable for culturing two kinds of cells at same time. The targeting capability was studied by injecting cell culture medium with FSCNO-DH nanoparticles into the device through inlet1 for exiting via

outlet1. As a result, the velocity of injected nanoparticle solution is homogeneous and nearly the same in the two chambers (Fig. 5c, ~0.6 mm s$^{-1}$) which is close to the blood flow speed in capillaries[35]. With this device, green fluorescence protein (GFP)-expressing aHUVECs and HUVECs were cultured in the two chambers for 12 and 24 h, respectively. As shown in Fig. 5d (where the fluid passes aHUVECs first) or Fig. 5e (where the fluid passes HUVECs first), the aHUVECs and HUVECs can form tube-like structure in the device after culturing for 12 h and particularly 24 h. When perfusing cell culture medium dissolved with FSCNO-DH nanoparticles for 3 h in the device, the FSCNO-DH nanoparticles can target only the aHUVECs regardless of the position of the two chambers with HUVECs and aHUVECs (Fig. 5d, e), confirming the aHUVECs-targeting capability of FSCNO-DH nanoparticles under dynamic flow conditions.

The stability of the FSCNO-DH nanoparticles was examined by incubating the FSCNO-DH nanoparticles in medium for 2 days and then investigating their capability of targeting the aHUVECs versus HUVECs under both static culture in Petric dish and dynamic culture in the aforementioned microfluidic device. As shown in Supplementary Fig. 19a for static culture or 19 b for dynamic culture of aHUVECs, the FSCNO-DH nanoparticles after incubating with cell culture medium can still target the aHUVECs regardless of the culturing conditions. This is probably due to the

strong interaction between the sulfonyl group (in fucoidan) and amino group (on ASCNO nanoparticles) by reducing the nucleophilicity and basicity of the amino group to form sulfonamide-like stable interaction[36,37], which ensures the stability of fucoidan on the nanoparticle surface. In order to show the fucoidan density is important for the targeting effect, FSCNO nanoparticles with reduced/low fucoidan (F$^{low}$SCNO nanoparticles) were prepared. As shown in Supplementary Fig. 20a, the zeta potential of the F$^{low}$SCNO-DH nanoparticles is increased to $-11.2 \pm 3.8$ mV if the amount of fucoidan during the nanoparticle preparation is reduced from 5 to 2 mg. Consequently, the targeting capability of the F$^{low}$SCNO-DH nanoparticles is weakened compared with the FSCNO-DH nanoparticles (Supplementary Fig. 20b versus Fig. 5b or Supplementary Fig. 19a). It is worth noting that P-selectins are expressed in all the tumor cells used in this study (Supplementary Fig. 21), suggesting the FSCNO-DH nanoparticles can also target tumor cells.

Next, we investigated the in vivo tumor-targeting capability of the FSCNO nanoparticles by encapsulating indocyanine green (ICG, an NIR dye) in them to form the ICG-laden FSCNO (FSCNO-I) nanoparticles for in vivo imaging. The encapsulation of ICG does not change the zeta potential of the FSCNO nanoparticles (Supplementary Fig. 7b versus Supplementary Fig. 5c). NCI/ADR-RES, A2780ADR, and OVCAR-8 cells ($1 \times 10^6$ cells per mouse) were injected subcutaneously at the left dorsal side of the upper hindlimb of mice. Comparatively, NCI/ADR-RES, A2780ADR and OVCAR-8 cells were co-injected with aHUVECs at the right side of the mice to study the targeting effects (N + EC, A + EC, and O + EC in short, $1 \times 10^6$ cancer cells and $2 \times 10^6$ aHUVECs per mouse, Fig. 6a). The mice were used for imaging when the tumor volume is ~100 mm³. At 3 or 6 h after intravenous injection, free ICG (50 µg per mouse in 100 µl of saline) is observable throughout the mice body without specific tumor accumulation (Fig. 6b). In contrast, enhanced fluorescence of ICG is observable in the tumor areas at both sides (labeled with L for left and R for right, respectively) for mice treated with the FSCNO-I nanoparticles (ICG: 50 µg per mouse in 100 µl of saline) at both 3 and 6 h after i.v. injection (Fig. 6b). To confirm the targeting capability of FSCNO nanoparticles, various organs and tumors from both sides were harvested for ex vivo imaging after sacrificing the mice at 6 h. The ICG fluorescence in tumors of ICG-treated mice is minimal (Fig. 6c). In contrast, strong fluorescence of ICG can be observed in the tumors of FSCNO-I nanoparticle-treated mice formed with all the three different types of cancer cells with/without aHUVECs (Fig. 6c). Importantly, the intensity of ICG fluorescence in tumors of the N+EC, A+EC, and O+EC groups of mice treated with the FSCNO-I nanoparticles are significantly stronger than that in tumors grown without aHUVECs or free ICG treatment (Fig. 6c, d), supporting the tumor vasculature-targeting capability of the FSCNO nanoparticles observed in vitro. The tumors were then cryosectioned and stained with CD31 antibody to examine the distribution of FSCNO-I nanoparticles via the ICG fluorescence. As shown in Fig. 6e–g and Supplementary Fig. 22, the FSCNO-I nanoparticles can target the tumor blood vessels formed from both the host endothelial cells and injected aHUVECs.

**In vivo antitumor capacity**. To check the antitumor capacity of FSCNO-DH nanoparticles, two drug-resistant (NCI/ADR-RES and A2780ADR) and one nondrug resistant (OVCAR-8) tumor models were used for in vivo studies. Similar to the in vivo imaging studies, co-injection of aHUVECs were included in each tumor model to understand how the tumor vasculature-targeting capability of the FSCNO-DH nanoparticles may affect their antitumor capacity in vivo. As shown in Fig. 7a, for NCI/ADR-

RES tumors, the co-injection of aHUVECs (With EC, the Saline group) does not significantly influence the tumor growth compared with injection of the NCI/ADR-RES cells alone (W/O EC, the Saline group) during the 7 weeks of experiments. Unlike the in vitro data that shows the combination of free DOX and HM (DOX+HM) may kill the cancer cells better than free DOX alone in Petri dish where bioavailability of the two agents is high (Fig. 4d), the DOX + HM treatment has minimal effects on the growth of tumors in vivo. This is not surprising because the NCI/ADR-RES tumor cells are multidrug-resistant and the bioavailability of the two agents in tumor is low when they applied systemically in their free form, which may be overcome by encapsulating them in the tumor-targeting FSCNO nanoparticles for delivery. Indeed, a decrease in tumor volume is observable for the treatment of FSCNO-DH nanoparticles and the treatment of FSCNO-D nanoparticles with laser irradiation (FSCNO-D+L), especially for the With EC condition (Fig. 7a). This is probably because more FSCNO nanoparticles could accumulate in the tumors grown from both cancer cells and aHUVECs for the condition as shown in Fig. 6c. Importantly, the treatment of FSCNO-DH nanoparticles combined with NIR irradiation (FSCNO-DH+L) gives the best antitumor efficacy. Since the mild photothermal effect has minimal impact on the viability of tumor cells (Supplementary Fig. 16a), this result further confirms that controlled release of P-gp inhibitor is crucial to overcome the multidrug-resistant capability of NCI/ADR-RES cells for killing them in vivo. Similarly, the FSCNO-DH+L treatment is more effective in killing tumors in the With EC group (Fig. 7a), confirming the importance of tumor vasculature targeting for enhancing the efficacy of nanoparticle-mediated cancer therapy.

Unlike NCI/ADR-RES tumor cells, co-injection with aHUVECs (With EC, the Saline group) could significantly enhance the tumor growth of the drug-resistant A2780ADR cells compared with injection of the A2780ADR cells alone (W/O EC, the Saline group, Fig. 7a and Supplementary Fig. 23a). These tumors grew much faster than the NCI/ADR-RES tumors and reached the maximum size allowed in our animal protocol after only 16 days. Similar to that observed for the NCI/ADR-RES tumors though, the DOX+HM treatment has minimal impact on the growth of the A2780ADR tumors with EC or W/O EC, both the FSCNO-DH and FSCNO-D+L treatments could inhibit the tumor growth, and the FSCNO-DH + L treatment gives the best therapeutic outcome among all the treatments (Fig. 7a). This further confirms the importance of controlled release of P-gp inhibitor in overcoming cancer resistance in vivo. Of note, there is no significant difference between the With EC and W/O EC groups for the FSCNO-DH+L treatment. This is mainly because the tumors in With EC group grow faster than that in the W/O EC group. In other words, the decrease in tumor volume for the With EC group after the FSCNO-DH+L treatment is much more than that for the W/O EC group (Supplementary Fig. 23b), demonstrating the importance of tumor vasculature targeting for enhancing the efficacy of nanoparticle-mediated cancer therapy.

Interestingly, co-injection of aHUVECs (With EC, the Saline group) does not significantly influence the nondrug resistant OVCAR-8 tumor growth compared with injection of the OVCAR-8 cells alone (W/O EC, the Saline group). As expected, for the nondrug-resistant OVCAR-8 tumors, there is no significant difference between the FSCNO-D+L and FSCNO-DH+L treatments (Fig. 7a), because the P-gp expression is minimal in these cells. It is also not surprising that the DOX+HM treatment has minimal impact on the growth of the A2780ADR tumors with EC or W/O EC. However, the FSCNO-DH treatment is more effective than the DOX+HM treatment in inhibiting the tumor growth, presumably because of the improved bioavailability of DOX delivered with the FSCNO-DH nanoparticles.

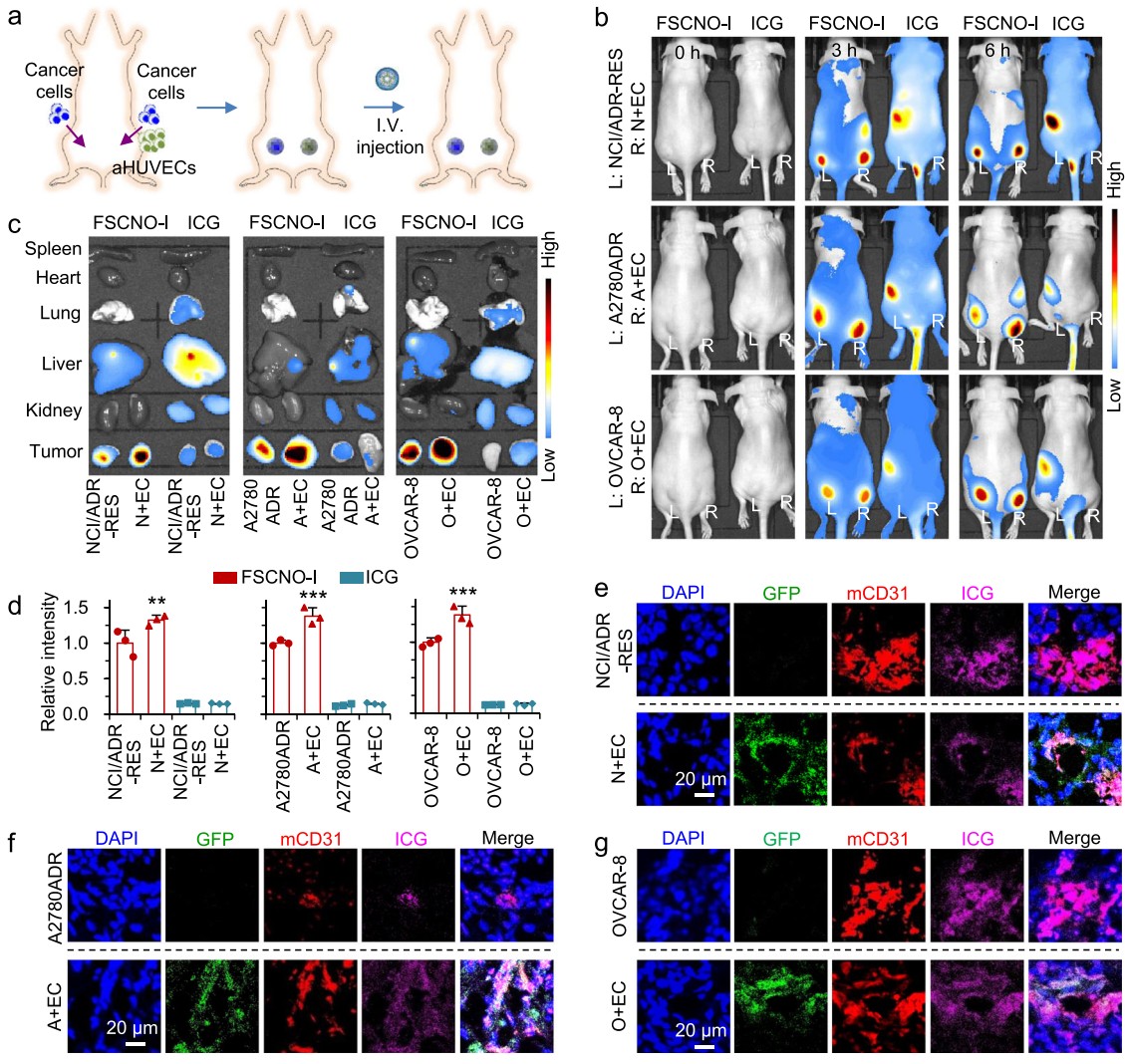

**Fig. 6 Tumor-targeting capability of FSCNO-DH nanoparticles. a** Schematic illustration of establishing tumors in mice by implanting NCI/ADR-RES, A2780ADR, or OVCAR-8 cells at the left dorsal side of the upper hindlimb of mice and the three types of cancer cells co-injected with aHUVECs at the right side of the same mice. **b** In vivo whole animal imaging of indocyanine green (ICG) fluorescence before and at different time points after intravenous (i.v.) injection via the tail vein in the form of free ICG and ICG-laden FSCNO (FSCNO-I) nanoparticles. Tumors on the left side (L) were formed with NCI/ADR-RES, A2780ADR or OVCAR-8 cells and tumors on the right side (R) were formed with co-injection of the respective cancer cells and aHUVECs (N+EC, A+EC, or O+EC). **c** Ex vivo imaging of ICG fluorescence in the tumors and critical normal organs collected from mice injected with FSCNO-I nanoparticles and free ICG. **d** Quantitative data on the intensity of ICG fluorescence in tumors collected from mice injected with FSCNO-I nanoparticles and free ICG. The data indicate the in vivo targeting capacity of the FSCNO-I nanoparticles. Moreover, there is a significantly enhanced accumulation of the FSCNO-I nanoparticles in N+EC, A+EC, and O+EC tumors compared with tumors formed with the respective cancer cells alone. Error bars represent ± s.d. ($n = 3$ mice for each group) and statistical significance was assessed by One-way ANOVA with Dunnett's post hoc analysis. **$p = 0.0096$, ***$p = 0.0004$, ***$p = 0.0002$ (left to right). **e–g** Fluorescence images of ICG, GFP, and CD31 in NCI/ADR-RES (**e**), A2780ADR (**f**), and OVCAR-8 (**g**) tumors collected from mice treated with FSCNO-I nanoparticles. The data indicate that aHUVECs are involved in tumor vasculature formation and the FSCNO-I nanoparticles can target tumor vasculature grown from both the native (i.e., host) endothelial cells or implanted aHUVECs.

Importantly, all the nanoparticle formulations are more effective in killing the tumors in the With EC than the W/O EC group, further demonstrating the importance of tumor vasculature targeting for enhancing the efficacy of nanoparticle-mediated cancer therapy. The aforementioned information was further confirmed with both the size and weight of the tumors from the various conditions and groups (Fig. 7b, c). Hematoxylin and eosin (H&E) stain shows extensive necrosis or shrinking in the tumors from the FSCNO-DH+L group while tumors from all the other groups are more viable (Fig. 7d).

Lastly, the body weights of mice with the FSCNO-DH+L treatment were stable during the whole observation period (Supplementary Fig. 24a). This may be due to the controlled drug

release inside tumors and minimal drug release outside tumors (Fig. 3e) as the body weight is stable for all mice treated with all the nanoparticle formulations (Supplementary Fig. 24a). Moreover, there is no evident damage to major organs collected from mice in the FSCNO-DH+L group as compared with the saline group, according to the H&E staining data (Supplementary Fig. 24b). In contrast, significantly decreased body weights of mice treated with DOX+HM were observed during the experiments, suggesting the DOX+HM treatment could cause systemic toxicity. This is confirmed by the H&E staining data showing cellular damage (vacuolization) in the heart and apoptotic hepatocytes in the liver of mice treated with DOX+HM. Interestingly, our previous studies show that free DOX

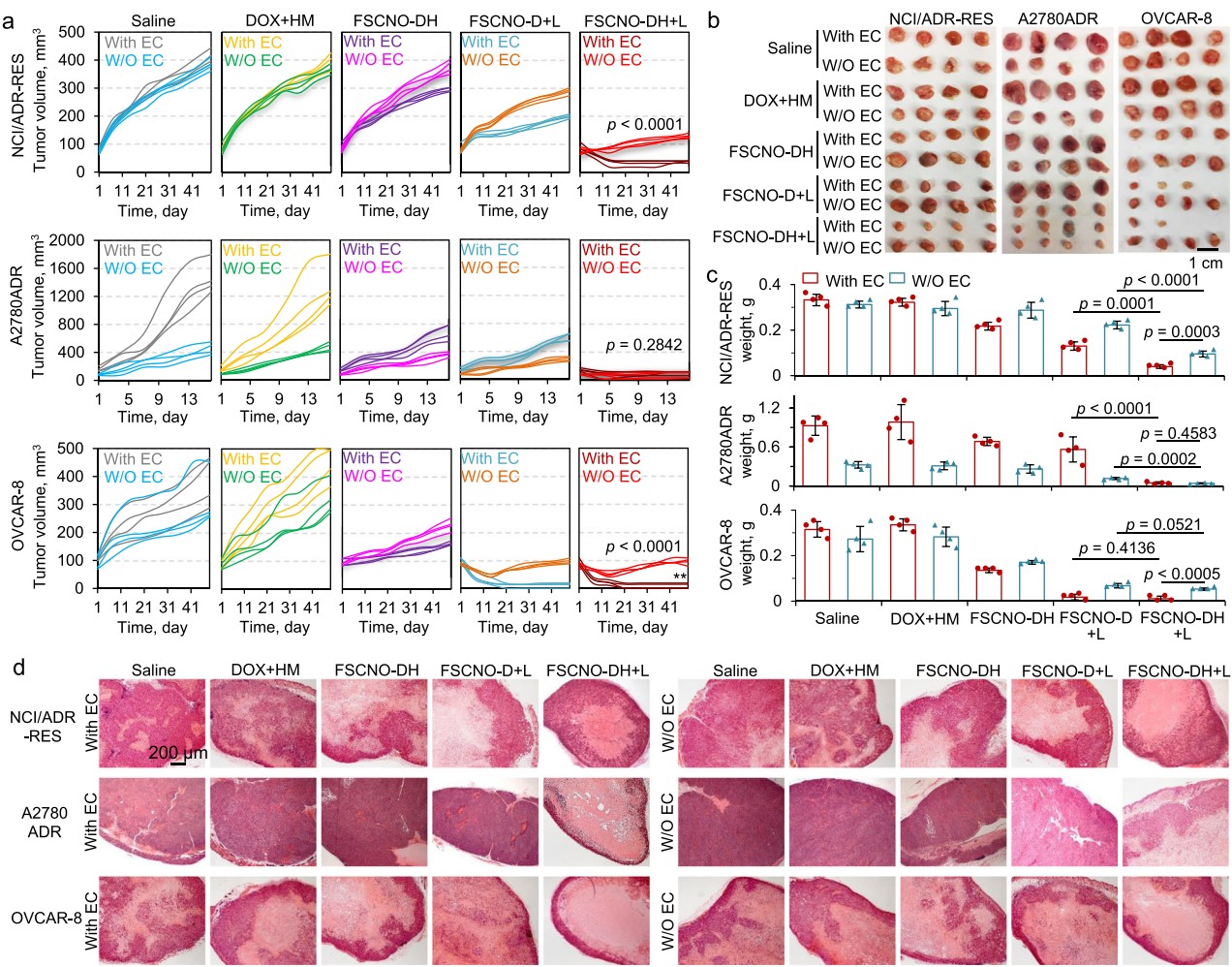

**Fig. 7 Capacity of the FSCNO-DH nanoparticles in overcoming drug resistance and destroying drug-resistant tumors in vivo. a–c** Growth curves (**a**), gross images (**b**), and weight (**c**) of NCI/ADR-RES, A2780ADR, and OVCAR-8 tumors with or without (W/O) co-injection of aHUVECs (With EC or W/O EC) in mice with various treatments, showing augmented antitumor efficacy of the FSCNO-DH+L treatments for all the three different types of tumors. This is due to the capability of the FSCNO-DH nanoparticles in targeting tumor vasculature and enhancing the targeted delivery of HM to inhibit the efflux pump in drug-resistant tumor when combined with the NIR laser irradiation. Error bars represent ± s.d. ($n = 4$ mice for each group) and statistical significance was assessed by unpaired two-sided Student $t$-test. **d** Representative images of hematoxylin and eosin (H&E) stained tumor tissues collected after sacrificing the mice on day 49 for the NCI/ADR-RES and OVCAR-8 groups and on day 16 for the A2780ADR group.

alone at the same dose as that used in this study does not induce evident decrease in body weight of mice[11,38], suggesting that the free HM co-injected with free DOX significantly increase the systemic toxicity of the free DOX. This might be because the injection of free HM could cause the dysfunction of the P-gp pumps in normal tissues. It is worth noting that the NIR laser irradiation alone could increase the temperature of tissues. Due to the superior photothermal effect of FCNO nanoparticles, an NIR laser power as low as 0.2 W cm$^{-2}$ was used for the in vivo studies in this work. This could avoid the potential damage to normal tissue (with no nanoparticles) surrounding the tumor at a higher power (e.g., 1 W cm$^{-2}$, Supplementary Fig. 25). Collectively, these data show the FSCNO-DH nanoparticles can improve not only the efficacy but also the safety of cancer chemotherapy via targeting and inhibiting the P-gp pumps.

## Discussion
Mesoporous silica nanoparticles have been used to encapsulate various hydrophilic and hydrophobic drugs[39–41]. In this study, the sequence of encapsulating DOX and HM in the FSCNO

nanoparticles was optimized. The encapsulation efficiency of HM is only ~24% if mixing HM with the FSCNO nanoparticles first, which is much lower than that (~69%) of the optimized method of mixing DOX with the nanoparticle first. This suggests that DOX might have some interactions with HM to enhance the encapsulation of HM inside the FSCNO nanoparticles. The encapsulation of DOX is probably because of the electrostatic interaction between DOX (positively charged) and silica (negatively charged) and the π–π stacking interaction between DOX and CNO. Although the exact mechanisms for the encapsulation of hydrophobic agents like HM in silica are still not well understood[42], one possible mechanism is the electrostatic interaction between the negatively charged silica and the local positive charge (e.g., the amino group) in the HM molecule. Another possible mechanism is the π-π stacking between the naphthalene-like structure in HM and the tetracene structure in DOX and CNO. This is supported with the FTIR data shown in Supplementary Fig. 26a. The aromatic bonds at 1582 cm$^{-1}$ can be observed only in the DOX, FSCNO-D, and FSCNO-DH groups, suggesting successful encapsulation of DOX in the nanoparticles. Similarly, methyl and carboxyl peaks at 1438 and 1407 cm$^{-1}$ are

observed mainly in the HM, FSCNO-H, and FSCNO-DH groups, confirming the existence of HM in the nanoparticles. Interestingly, multiple peaks associated with DOX and HM disappear or reduce after they are encapsulated inside the nanoparticles (i.e., FSCNO-D, FSCNO-H, and FSCNO-DH). This might be due to the π–π stacking between FSCNO nanoparticles and DOX/HM that decreases the molecule vibrations. Because the disappeared or reduced peaks associated with DOX or HM are not evidently affected if they are simply mixed with the FSCNO nanoparticles for the measurements (i.e., FSCNO+D, FSCNO+H, or FSCNO+D +H, Supplementary Fig. 26b). Moreover, the π–π stacking between FSCNO nanoparticles and DOX is supported by the fluorescence spectra shown in Fig. 2g. This is consistent with the literature[43,44], showing the DOX fluorescence intensity decreases after it is encapsulated in nanoparticles that have π-stacking interactions with DOX, compared with free DOX.

For controlled drug release using NIR laser, our data show that the triggered drug releases from silica-fullerene (note: fullerene is a carbon nanomaterial as with nano-onion) hybrid nanoparticles is mainly because of the NIR laser-induced temperature increase in a previous study[38]. Similarly, the temperature in the FSCNO-DH nanoparticles should increase during the laser irradiation. As a result, the Brownian motion of all molecules is increased, which may destabilize the electrostatic and π–π stacking interactions between HM/DOX and the surface within the nanoparticles. This may free the HM and DOX for them to diffuse out of the nanoparticles under concentration gradient.

When controlling the drug release, HM and DOX are triggered to release simultaneously from the FSCNO nanoparticles during the NIR irradiation. Nonetheless, HM can quickly, selectively, and potently inhibit the P-gp function. As a result, although some DOX may be pumped out of cells before the function of all P-gps is inhibited, it could re-enter the cells after the HM binds with all the P-gps quickly. To support this, the multidrug resistance cells (NCI/ADR-RES and A2780ADR) were either pre-treated with HM for 30 min (HM-DOX) or without any HM pre-treatment (HM+DOX) before incubating them with both HM and DOX for 3 h to examine the DOX uptake by the cells. As shown in Supplementary Fig. 27a, b for NCI/ADR-RES and A2780ADR cells, respectively, the intracellular fluorescence intensity of DOX for the HM-DOX treatment is similar to that for the HM+DOX treatment. Similarly, the cytotoxicity of the HM-DOX treatment is not significantly different from that of the HM+DOX treatment (Supplementary Fig. 28a,b). These data confirm that releasing HM and DOX simultaneously from the FSCNO nanoparticles is of significance for overcoming cancer drug resistance.

Lastly, the purpose of co-injecting aHUVECs with tumors cells was to mimic the condition of tumors with various degree of P-selectin expression in this study. This is also helpful to confirm the importance of the P-selectin targeting capability of the FSCNO nanoparticles for enhancing cancer therapy. Importantly, the FSCNO-DH+L treatment can still significantly inhibit the growth of the NCI/ADR-RES, A2780ADR, and OVCAR-8 tumors without co-injection of the aHUVECs, compared with the treatments of free drug, FSCNO-D+L, and FSCNO-DH. In other words, the FSCNO-DH nanoparticles with laser irradiation can efficiently inhibit the growth of multidrug-resistant tumors in general and is even more effective for tumors with high P-selectin expression or ample blood vessels.

In summary, we synthesized a fucoidan-decorated silica and CNO hybrid (FSCNO) nanoparticle for targeting the tumor vasculature and P-gp overexpressed on drug-resistant cancer cells. We further revealed the superior light absorption at the NIR region of CNO (in comparison to the widely studied CNT and GO). As a result, controlled release of P-gp inhibitors and chemotherapeutic drugs can be achieved for the FSCNO

nanoparticles under NIR irradiation at a very low power. The FSCNO-DH nanoparticles can precisely compromise the function of P-gp pumps and overcome the drug resistance of multidrug-resistant NCI/ADR-RES and A2780ADR cancer cells both in vitro and in vivo. The tumor vasculature-targeting capability of the nanoparticles via the binding between fucoidan and P-selectin overexpressed in tumor vasculature is confirmed by three different models: static culture of activated human endothelial cells in Petri dish, dynamic culture of activated human endothelial cells in microfluidic device, and human tumors in mice. In particular, the microfluidic model developed in this study may be valuable for mimicking the in vivo perfusion in blood vessels to examine the targeting capability of nanoparticles. Ultimately, the chemotherapeutic agent (DOX) and P-gp inhibitor (HM)-laden FSCNO nanoparticles exhibit superior safety and efficacy for therapy of two types of multidrug-resistant tumors and one type of nondrug-resistant tumors without evident side effects. This study may lay the foundation of developing a promising nanotechnology-based strategy for combating cancer multidrug resistance.

## Methods

**Materials**. CNOs were purchased from Graphitic Nano Onions LLC (IN, USA). CNTs were purchased from NanoLab, Inc (Waltham, MA, USA). GOs were purchased from Graphenea Inc (Cambridge, MA, USA). Hexanol, cyclohexane, triton X-100, toluene, TEOS, APTMS, fucoidan, and ICG were purchased from Sigma (St. Louis, MO, USA). DOX was purchased from LC laboratories (Woburn, MA, USA). HM30181A was purchased from MedChemExpress LLC (Monmouth Junction, NJ, USA). The CCK-8 cell proliferation reagent was purchased from Dojindo Molecular Technologies (Rockville, MD, USA). Fetal bovine serum (FBS) and penicillin/streptomycin were purchased from Invitrogen (Carlsbad, CA, USA). The RPMI 1640 and EMEM cell culture media were purchased from ATCC (Manassas, VA, USA). All other chemicals were purchased from Sigma unless specifically mentioned otherwise.

**Preparation of nanoparticles**. For SCNO nanoparticles, CNOs suspended in benzene was mixed with hexanol (5 ml), triton X-100 (1.7 ml), and cyclohexane (2 ml). Then, 60 μl of ammonium hydroxide (28 wt%) and 100 μl of TEOS were added into the mixture and stirred at 800 rpm for 12 h at room temperature using a small stir bar. The reaction was then stopped by adding 30 ml of ethanol. The SCNO nanoparticles were collected by centrifuging at $13,800 \times g$ for 10 min and purified by washing with 2 ml of ethanol and water for two times in each liquid. Next, 20 μl of APTMS in 3 ml of ethanol was mixed with SCNO nanoparticles and reacted for 12 h to obtain the ASCNO nanoparticles after washed with 2 ml of ethanol and water for two times. Lastly, the FSCNO nanoparticles were prepared by mixing the ASCNO nanoparticles with 5 or 2 ml of fucoidan solution (1 mg ml$^{-1}$ in deionized water) and stirred for 12 h at room temperature. For synthesizing the FSCNT and FSGO nanoparticles, the same procedure was used except that the CNOs were replaced with CNTs and GOs.

**Characterization of nanoparticles**. The size distribution and zeta potential of nanoparticles (1 mg ml$^{-1}$ in deionized water) was measured using a Malvern Zetasizer Nano ZS instrument (Cambridge, UK). To further confirm the size and morphology of the nanoparticles, the FSCNO nanoparticles were imaged in uranyl acetate solution (2%, w w$^{-1}$) using an FEI (Moorestown, NJ, USA) Tecnai G2 Spirit transmission electron microscope. The CNOs were further imaged using a Tecnai G2 F20 U-TWIN high-resolution transmission electron microscope. For FSCNT and FSGO nanoparticles, the nanoparticle solutions were dropped on a freshly cleaved mica grid and dried for 3 h at room temperature. Samples were imaged with a FEI NOVA Nano400 scanning electron microscope after coated with a thin film of Au onto the nanoparticles.

FTIR spectroscopy analysis of the SCNO, SCNT, and SGO nanoparticles was conducted using a Perkin Elmer (Waltham, MA, USA) Spectrum 100 FTIR spectrometer. Dried samples of the various nanoparticles were grounded in an agate mortar, mixed with KBr at a ratio of ~1:80 (nanoparticle: KBr) in weight, and pressed into small discs at 10 tons for 5 min. The similar procedure was used for the FTIR characterizations of DOX, HM, FSCNO-D, FSCNO-H, FSCNO-DH and the mixtures of DOX and HM, FSCNO and DOX, FSCNO and HM, and FSCNO with both drugs.

The N$_2$ sorption measurements of the FSCNO nanoparticles were determined by an automated nitrogen adsorption analyzer (Quantachrome Instruments, Boynton Beach, FL) at 77 K. Prior to the sorption measurements, FSCNO nanoparticles were pre-treated under high vacuum at 80 °C for 12 h.

**In vitro photothermal measurement**. FSCNO, FSCNT, and FSGO nanoparticles in deionized water with different concentrations (0.5, 0.25, 0.12, and 0.06 mg ml$^{-1}$) were prepared and irradiated with NIR laser (800 nm, 1 W cm$^{-2}$) for 2 min. The temperatures of nanoparticle solutions were recorded using thermocouples.

**Encapsulation of HM and DOX and in vitro drug release**. To encapsulate the therapeutic agents, DOX was first mixed with FSCNO nanoparticles at a feeding ratio of 1:20 (DOX: nanoparticles in weight) in deionized water for 24 h according to previous studies[11,45,46]. Then, the nanoparticles were collected by centrifuging at 13,800 × g for 10 min and further mixed with HM dissolved in DMSO (1 mg ml$^{-1}$) at a feeding ratio of 1:20 (HM: nanoparticles) in deionized water for 24 h. The following equation was used to calculate the encapsulation efficiency (EE) of DOX and HM:

$$EE = W_{encapsulated} \times W_{fed}^{-1} \times 100\% \qquad (1)$$

where $W_{encapsulated}$ represents the amount (in weight) of DOX and HM encapsulated into nanoparticles and $W_{fed}$ is the initial total amount of DOX and HM fed for encapsulation. The amount of DOX and HM was measured using spectrophotometry at absorbance of 485 and 350 nm, respectively.

Similarly, ICG was mixed with FSCNO nanoparticles at a feeding ratio of 1:20 (ICG: nanoparticles in weight) in deionized water for 24 h for encapsulation. Afterward, the sample was washed with DI water for three times to remove any nonencapsulated ICG for in vivo imaging.

To investigate drug release, FSCNO-DH nanoparticles were dissolved in PBS (1.5 ml, at pH 7.4) and placed in a shaker (ThermoFisher Scientific Inc., Waltham, USA) at 110 rpm and 37 °C. For measurement at various timepoints, the nanoparticle solution was centrifuged at 13,800 × g to obtain the supernatant. The released DOX or HM in the supernatant was analyzed using its absorbance at 485 or 350 nm measurement using spectrophotometry for DOX and HM, respectively.

**Cell culture and in vitro cell viability**. HUVECs were cultured in Endothelial Cell Medium (ScienCell Research Laboratories, Inc) consisting of basal medium, 10% FBS, 1% endothelial cell growth supplement, and 1% antibiotic solution at 37 °C in a humidified 5% CO$_2$ incubator. The HUVECs were activated by adding human TNF-α (50 ng ml$^{-1}$, ab9642, Abcam) into their medium to culture for ~5 h. Human multidrug-resistant NCI/ADR-RES cancer cells (ATCC) were cultured in EMEM supplemented with 10% FBS and 1% penicillin/streptomycin at 37 °C in a humidified 5% CO$_2$ incubator. Human multidrug-resistant A2780ADR cancer cells (Sigma–Aldrich) and OVCAR-8 cancer cells (ATCC) were cultured in RPMI 1640 medium supplemented with 10% FBS and 1% penicillin/streptomycin. To maintain the multidrug-resistant property of NCI/ADR-RES and A2780ADR cells, 1 μg ml$^{-1}$ of DOX were added into their respective media during culturing. For in vitro anticancer studies, the NCI/ADR-RES, A2780ADR, and OVCAR-8 cells were incubated with various drug formulations for 24 h. Free HM were dissolved in DMSO (1 mg ml$^{-1}$) and mixed with medium based on the experimental HM doses. During the laser irradiation, the medium containing drug formulations was replaced with pure medium at 12 h and the cells were then irradiated laser with different power (0.05, 0.2, 0.5, and 1 W cm$^{-2}$) for 1 min. Cells were further cultured with medium containing various drug formulations for 12 h. The cell viability was measured with the CCK-8 assay according to the instructions given by the manufacturer.

**In vitro imaging**. Single cancer cells were collected and cultured on the collagen-coated cover glasses in six-well plates at a density of 2 × 10$^5$ cells per well at 37 °C for 12 h. For cellular uptake of DOX, the medium in each well was then replaced with 2 ml of fresh medium containing different drug formulations and culture for 6 h. Cancer cells were then fixed with 4% paraformaldehyde (PFA) at room temperature for 15 min and incubated with 75 nM of LysoTracker Green DND-99 (Life Technologies, Waltham, MA, USA) for 30 min at room temperature to stain the endosomes/lysosomes. The cell nuclei were stained with 4'6-diamidino-2-phenylindole (DAPI, 300 nM) for 10 min at room temperature and washed three times with PBS before imaging. For P-selectin imaging, HUVECs or aHUVECs cultured on collagen-coated cover glasses were fixed with 4% PFA for 15 min at room temperature. After washed with 2 ml of PBS for three times, cells were incubated in 3% BSA in 2 ml of PBS at room temperature for 3 h to block potential nonspecific binding. Then, the fixed cells were incubated with P-selectin antibody (cat: 701257, ThermoFisher) overnight at 4 °C at the dilution ratio of 1:100. Unbounded antibody was carefully removed by washing with 500 μl of PBS for three times. Cells were then incubated with secondary antibody (ThermoFisher) at the dilution ratio of 1:200 in PBS with 1% BSA at room temperature for 1 h and then washed three times with 500 μl of PBS. For the binding studies with 2D-cultured cells, aHUVECs cultured on collagen-coated cover glasses were put on ice for 3 h to minimize their metabolic and uptake activities. Then, medium was replaced with 2 ml of cold fresh medium containing SCNO-DH or FACNO-DH nanoparticles (10 μg ml$^{-1}$ for DOX and 5 μg ml$^{-1}$ for HM) and incubated for 6 h at the cold temperature. The aHUVECs were then fixed with 4% PFA at the cold temperature and immunostained for P-selectin as aforementioned. The cover glass attached with cells was mounted onto a glass slide with anti-fade mounting medium (Vector Laboratories Burlingame, CA, USA) for examination using a Zeiss LSM 700 confocal

microscope. For P-gp imaging, the blocked cancer cells were incubated with Anti-P-Glycoprotein antibody (P7965, Sigma–Aldrich) overnight at 4 °C at the dilution ratio of 1:200 and treated as above mentioned.

**Flow cytometry analysis**. The NCI/ADR-RES, A2780ADR, and OVCAR-8 cells were incubated in medium containing various drug formulations for 6 h. After irradiated with laser for 1 min at 0.2 W cm$^{-2}$ for the FSCNO-DH+L group, all the samples were washed twice with PBS, detached, and fixed with 4% PFA for 20 min at room temperature. The fixed cells were then washed twice and analyzed using a BD (Franklin Lakes, NJ, USA) Accuri C6 flow cytometer.

**Fabrication of microfluidic perfusion device**. To fabricate the microfluidic device, the microfluidic channel and chamber designs were created using AutoCAD and printed onto a plastic sheet to create a plastic shadow mask (CAD/Art Services, Inc. Bandon, OR, USA). The 100 mm silicon wafer (University Wafer, South Boston, MA, USA) were first cleaned using methanol, acetone, and isopropyl alcohol (IPA) and dried using nitrogen gas. SU-8 2100 photoresist (Kayaku Advanced Materials, Westborough, MA, USA) was then deposited on the wafer, which was spun to create a thin (300 μm) layer of photoresist on the wafer. After spinning, a soft bake was performed. The wafer with photoresist and shadow mask were then placed in an EVG 620 mask aligner. The photoresist on the wafer was exposed to UV at 400 mJ cm$^{-2}$ through the channel or chamber designs printed in the shadow mask, followed by a post exposure bake. The wafer with photoresist was developed with SU-8 developer (Kayaku Advanced Materials, Inc) for 20 min followed by an IPA and water rinse and was dried using nitrogen gas. A hard bake was performed to create the clean and dry master or mold for making the microfluidic device by soft lithography. Spin speed, baking times, baking temperatures, and exposure energy was determined using the product processing guidelines.

To create the microfluidic device, a 1:10 mixture of PDMS elastomer curing agent to elastomer base (Ellsworth Adhesives, Germantown, WI, USA) was poured over the mold and baked at 65 °C for at least 2 h. The PDMS was then carefully peeled off the mold and plasma-treated using a PDC-32G plasma cleaner (Harrick Plasma, Ithaca, NY, USA) at 18 W and 27 Pa for 2 min. Afterward, the top and bottom pieces of the devices were wetted with methanol, aligned under microscope, and bound together. The PDMS devices were then immediately placed in a 65 °C oven to evaporate methanol and baked in the 65 °C oven for three days for further use.

For imaging of cells in the microfluidic device, GFP-expressed HUVECs or aHUVECs were injected into the chambers through inlet2 or inlet3 at a density of 20,000 cells per chamber. After culturing for 12 or 24 h, fresh medium containing FSCNO-DH nanoparticles (10 μg ml$^{-1}$ for DOX and 5 μg ml$^{-1}$ for HM) was continuously injected into the device through inlet1 at 0.2 ml min$^{-1}$ for 3 h. Then, cells were stained with DAPI for 10 min and washed with PBS by flowing it from inlet1 at 0.2 ml min$^{-1}$ for another 10 min. The cells were then examined directly in the devices using a Zeiss LSM 700 confocal microscope.

**Computational modeling of velocity distribution in the microfluidic device**. The COMSOL (Burlington, MA, USA) Multiphysics (v5.2) (Fluid Flow module) was used to calculate the flow field in the microfluidic perfusion device. The single-phase and laminar flow in the chambers and channels was considered to be incompressible. The governing equation for the steady-state fluid flow was as follows:

$$\rho(\boldsymbol{V} \cdot \nabla)\boldsymbol{V} = -\nabla p + \mu\nabla^2\boldsymbol{V} + \boldsymbol{F_b} \qquad (2)$$

where $\rho$, $\boldsymbol{V}$, $p$, $\mu$, and, $\boldsymbol{F_b}$ represent density (1000 kg m$^{-3}$), velocity, pressure, viscosity (1 mPa s), and body force (0), respectively. The nonslip boundary condition ($\boldsymbol{n} \cdot \boldsymbol{V} = 0$, where $\boldsymbol{n}$ is outward normal of the boundary) was applied on the microchannel or chambers walls. The normal inflow speed at all the inlets 1, 2, and 3 was 3 mm s$^{-1}$ and the pressure of the outlet was 0. The parameters in bold were vectors and non-bold parameters were scalars.

**Animals and xenograft tumors**. Athymic female NU/NU nude mice (6-week old) were purchased from Charles River (Wilmington, MA, USA) and maintained on a 16:8 h of light-dark cycle under temperatures of ~18–23 °C with 40–60% humidity. All procedures for animal use were approved by the Institutional Animal Care and Use Committee (IACUC) at the University of Maryland, College Park, MD. The animal protocols are compliant with all relevant ethical regulations. To obtain xenografts in the nude mice, NCI/ADR-RES, A2780ADR, and OVCAR-8 cells were suspended in a mixture (1:1) of 1× PBS and Matrigel and injected subcutaneously at the left dorsal side of the upper hindlimb of mice (1 × 10$^6$ cells per mouse). In parallel, NCI/ADR-RES, A2780ADR, and OVCAR-8 cells were co-injected with aHUVECs (ratio of cancer cells to aHUVECs was 1:1, 2 × 10$^6$ cells per mouse) at the right dorsal side of the upper hind limb of each 7-week-old mouse.

**In vivo imaging and biodistribution**. When the tumor volume reached ~100 mm$^3$ in long diameter, the mice were intravenously injected with 100 μl of saline, 50 μg of free ICG dissolved in 100 μl of saline, 50 μg of ICG encapsulated in encapsulated in FSCNO nanoparticles dissolved in 100 μl of saline via the tail vein. All the mice

were imaged at 0 h (right before injection), 3 h, and 6 h after injection, using a PerkinElmer (Waltham, MA, USA) IVIS instrument with excitation at 780 nm and an 831 nm filter to collect the fluorescence emission of ICG. For ex vivo imaging, the mice were sacrificed at 6 h and the major organs (liver, kidney, lung, spleen, and heart) and tumors were collected for further ex vivo fluorescence imaging of ICG in the same way as aforementioned for whole body imaging.

For staining of mouse CD31 (R&D Systems, AF3628), tumor tissues were collected and frozen using Tissue-Tek (Sakura Finetek, Torrance, CA, USA) O.C.T. Compound and Cryomold at −80 °C. Tumor tissues were cut into 10-μm thick slices and incubated in 3% BSA and 0.1% TritonX-100 in 1× PBS at room temperature for 1 h to block potential nonspecific binding. After incubated with mouse CD31 antibody (1:200 dilution) at room temperature for 2 h, the samples were washed three times with PBS and incubated in the dark at room temperature for 1 h with Alexa Fluor 594 labeled secondary antibody (ThermoFisher) diluted in 1% BSA in 1× PBS (1:50 dilution). Afterward, the slides were examined using a Zeiss LSM 700 confocal microscope.

**In vivo antitumor efficacy and safety**. After tumors volume reached ~100 mm$^3$, mice were randomly allocated to each group and treated with 100 μl of saline, free DOX + HM, FSCNO-D nanoparticles with NIR laser irradiation (FSCNO-D + L), and FSCNO-DH nanoparticles without or with NIR laser irradiation (FSCNO-DH + L) (n = 4). All the drug formulations were injected in 100 μl of saline, except for free HM that was dissolved in DMSO and mixed with saline at a ratio of 1:10 (DMSO: saline, 200 μl per mouse) in ultrasonic water baths for 15 min. The dose of DOX and HM for all the formulations with the drug was 2.5 and 1 mg kg$^{-1}$ body weight, respectively. At 12 h after injecting various drug formulations, tumors were irradiated with NIR laser at the power of 0.2 W cm$^{-2}$ for 2 min if needed. Of note, the laser irradiation was paused after 1 min and continued after tumor was cooled back to room temperature in ~5 min measured with FLIR E6 Compact Thermal Imaging Camera (FLIR Systems, Inc. Arlington, VA, USA). These treatments were conducted twice on days 1 and 7. The tumor volume (V) was calculated as: V = L × W$^2$ × 0.5, where L is long diameter and W is short diameter determined using a caliper. The mice injected with A2780ADR cells were euthanized on day 16 while NCI/ADR-RES and OVCAR-8 injected mice were euthanized on day 49. Tumors, livers, lungs, hearts, spleens, and kidneys from different groups were collected and fixed in 4% PFA for H&E staining.

**Statistics and reproducibility**. All data are reported as mean ± standard deviation (s.d.) from at least three independent runs. The Student t-test were used to assess two groups of independent samples, assuming equal variance with no samples being excluded from analysis using Microsoft Excel. One-way ANOVA with Dunnett's post hoc analysis was used for multiple comparison (i.e., when more than two groups were compared) using GraphPad Prism 8. For graphs showing representative data, n ≥ 3 independent experiments for Fig. 2b, c; n = 3 independent experiments for Figs. 4a–c; 5a, b, d, e; 6e-g; n = 4 independent experiments for Fig. 7d and n = 3 independent experiments for Supplementary Figs. 2; 3; 8; 10; 11; 12; 13; 19; 20b; 21; 22; and n = 4 independent experiments for Supplementary Fig. 24b. In all cases, a p value less than 0.05 was considered to be statistically significant.

**Reporting summary**. Further information on research design is available in the Nature Research Reporting Summary linked to this article.

## Data availability

Data supporting the findings of this study are available within the paper and its Supplementary Information files. The data that support the findings of this study are available from the corresponding author on request. Source data are provided with this paper.

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

## Acknowledgements

This work was supported by grants from the National Institutes of Health (NIH R01CA206366 and R01CA243023).

## Author contributions

H.W. and X.H. conceived the project; H.W., Y.L., and X.H. analyzed the data and wrote the manuscript; H.W. conducted all the experiments with help from Y.L., Y.Y., A.M.W., B.J., J.X., J.Z., W.S., and S.S.; Y.Z. conducted the computational modeling studies; X.L. edited the manuscript; and all authors approved the manuscript.

## Competing interests

The authors declare no competing interests.
