## [Peer Review File · Nature Communications]

REVIEWER COMMENTS

Reviewer #1 (Remarks to the Author): expert in nanoparticles

The paper reports the construct of a fucoidan-decorated silica-carbon

41 nano-onion (FSCNO) hybrid nanoparticle for targeting tumor vasculature and P-glycoprotein to overcome cancer drug resistance. Different *in vitro* and *in vivo* models have been used to investigate the targeting efficacy, very interesting and promising results were obtained. However, several key questions need to be addressed.

(1) Fucoidan is adsorbed on the nanoparticle surface through electrostatic interactions. How stable are this surface coating *in vitro* and *in vivo*? As it is known that electrostatic interaction is not strong enough to ensure the stability of surface coating, when in salt or cell culture medium, the surface coating could fall. Also, what's the density of Fucoidan on the particle surface? Does the Fucoidan density on the surface affect their tumor vasculature targeting?

(2) Drug loading: as the FSCNO nanoparticles have a CNO core and silica shell coated with Fucoidan. What's the mechanism of drug (DOX and HM) loading? Where are the drugs loaded? On the particle surface? DOX, a water soluble drug, was firstly loaded, what's the interaction between DOX and the particle? Does the DOX coating cover the Fucoidan? Also, following the loading of DOX, then HM in DMSO was further mixed with the nanoparticles, is the HM also on the particle surface covering the DOX? as HM is hydrophobic, what forces or interactions keep the hydrophobic HM on the particle surface then? What's the drug loading (the ratio of drug to the drug-loaded nanoparticle mass)? The layer DOX and HM should have covered the Fucoidan, how these coatings affect the targeting effect of Fucoidan?

(3) Page 9, line 219, "This is probably due to the superior photothermal effect of FSCNO nanoparticles, which could effectively trigger the release of HM to inhibit the P-gp and enhance the retention of DOX in the drug resistant cells for the observed effective destruction of the cells" How the NIR irradiation affect drug release? What's the trigger release mechanism? Were both the drugs capable of trigger release? Does the laser power affect the integrity of the nanoparticles?

(4) For *in vivo* targeting, the surface property of the nanoparticles plays a critical role. What's the surface property of the nanoparticles after drug loading? Are they a mixture of DOX, HM and Fucoidan? How are they distributed on the nanoparticle surface? and what are their densities on the

particle surface? HM is very hydrophobic, which could affect the circulation of nanoparticles adversely? As hydrophobic nanoparticles could be quickly cleared from the circulation once injected.

(5) In in vivo experiments, the authors used indocyanine green loaded nanoparticles for investigating tumor targeting capability? How was ICG loaded? Are they also loaded on the particle surface through absorption? Then the surface property of ICG loaded nanoparticles should be very different from DOX and HM loaded nanoparticles, as ICG has very different property from HM.

(6) For in vivo antitumor capacity experiments, it can be seen that FSCNO-DH had very limited effect in inhibiting the tumor growth for all the three cells, so it would be very helpful to have a control experiment of FSCNO + laser to demonstrate that just targeted photothermal treatment is sufficient.

Reviewer #2 (Remarks to the Author): expert in nano-onions

The manuscript of Wang et al. reports on the development of fucoidan-decorated silica and CNO hybrid (FSCNO) nanoparticle and their potential application for selective targeting of tumor vasculature and anticancer effect, thanks to the co-deliver of a P-gp inhibitor and a chemotherapeutic drug.

This is a very interesting paper, which clearly shows the potential biomedical applications of carbon-based nanomaterials. The studies prove the selective targeting of tumor vasculature on three different model: a cell culture, a microfluidic dynamic culture and finally in vivo on a tumoral mice model.

The delivery of P-gp inhibitors directly to multidrug-resistant cancer cells is a clever idea that has allowed overcoming drug resistance both in vitro and in vivo.

Moreover, the authors demonstrated the capability of these new smart nanohybrids to release the anti cancer drug upon internal trigger into the tumor area, by using the high light absorption of carbon nano-onions in the NIR. The manuscript is well written, the subject is new and the data are solid.

The scope of the problem tackled by the paper significant enough to warrant a Nature publication.

I believe that this paper matches the quality criteria requested for publication in Nature Communications and therefore suggest its publication after minor corrections.

List of comments:

1. While it is a clever idea to coat the CNOs with TEOS and APTMS to improve their solubility and loading capacity and the Silica-CNO nanoparticles (SCNO) produced look uniform in the TEM images (Supp. Fig. 2), the TEM images of the starting material shown in Figure 2b are not convincing of CNO. The author comment “the nano-onion structure (~50 nm in diameter) can be observed only at the edge of the aggregates at higher magnification (right, Fig. 2b). A better HRTEM image of individual CNO should replace Fig. 2b. It is also a little confusing the following statement “Interestingly, the hybrid SCNO nanoparticles are smaller than the CNOs”. The authors are invited to carefully revise.

2. There are typos and errors in the paper, the reviewer recommends proofreading the paper an additional time missing units on line 129 & 130. Is line 287 is supposed to say ' Similar to in vitro imaging', rather than 'Similar to in vivo imaging' ? Error on line 311, etc.

3. The DLS (Fig. 2d) should be given as a line plot instead of a scatter plot, to better see the curve,

4. The UV-Vis solvent is not given

Reviewer #3 (Remarks to the Author): expert in p-selectin targeted nanoparticles

In this article, the authors reported a fucoidan-decorated silica-carbon nano-onion (FSCNO) hybrid nanoparticle that co-deliver Pgp inhibitor and chemotherapeutic drug (DOX) for precisely targeting tumor vasculature by specifically releasing P-gp inhibitor and anticancer drug into tumor cells. The carbon nano-onion was used to trigger the drug release via NIR and Silica surface-decorated with fucoidan (F) was used as a target to specifically bind to P-selectin that is overexpressed on tumor vasculature in this nanoplatform (FSCNO). DOX is a chemotherapeutic drug.

The most contribution in this paper is that the authors try to achieve an effective inhibition on P-gp vis using P-gp inhibitor to reduce multi-drug resistant. The functions of the components used for the synthesis of the nanoparticles are well known and the paper indeed made a well detailed in-vitro and in-vivo experiments to prove the FSCNO nanoparticle with the above-mentioned functions. The main claim in this paper is emphasized to control the release of P-gp inhibitor via NIR for effectively inhibiting the P-gp. However, what is the novelty I am thinking? Combination of fucoidan and P-gp release to exhibit the targeting of tumor vasculature and P-gp inhibition indeed contains somewhat important contribution and novelty but is not so highly obvious enough. Some questions and concerns about the paper are listed below.

1. The particle size of FSCNO is very small about 20-25 nm and Transmission electron microscopy (TEM) images of the CNOs in Figure 2 (b) and FSCNO nanoparticles (c) showed its morphology looks like a solid particle (not hollow structure) with a very small capacity to load both Pgp inhibitor and chemotherapeutic drug (DOX) inside the nanoparticle. So Please provide the loading capacity and nature release of both drugs.
2. Another question is how both hydrophobic HM and doxorubicin hydrochloride (??_) were encapsulated in the hydrophobic CNO since HM is hydrophobic and doxorubicin hydrochloride tend to be dissolved in hydrophilic solvent like water so both drugs are very difficult physicochemical characters. According the experimental illustration in this paper, they mentioned that DOX first mixed with FSCNO nanoparticles at a feeding ratio of 1:20 (DOX : nanoparticles in weight) in deionized water for 24 h. Then, the nanoparticles were collected and further mixed with HM dissolved in DMSO (1 mg/ml) (HM : nanoparticles) in deionized water for 24 h. Therefore, I think that the authors should provide more explanation to illustrate how both drugs were encapsulated and loading capacity of HM and Dox.
3. Next, the control release for both Pgp inhibitor and chemotherapeutic drug (DOX) from the FSCNO nanoparticle is very important. Figure 3(e) only showed both DOX and HM are simultaneously released only under the trigger with near infrared (NIR) laser irradiation (0.5 W/cm²) for 1 min. However, how much FSCNO nanoparticles can be uptaken by the cancer cells since the FSCNO nanoparticles have a very high negative-charged surface, which is usually not favorable for cell uptake. They should provide the data measured by Flow cytometry.
4. Another more important question is that as the FSCNO was uptaken by the cancer cells, how the release of P-gp inhibitor and chemotherapeutic drug from FSCON nanoparticles can be effectively controlled ?? Here, it only showed both drugs are released simultaneously in Figure 3(e). However, it is better to release HM first to inhibit the P-gp and then the released DOX can be avoided to be pumped out. Please explained it in detail. Otherwise, The FSCNO nanoparticles can not achieve direct inhibition of the drug efflux pumps of DOX for overcoming the multidrug resistance of cancer.
5. Figure 3(e) showed the drug release triggered laser at 0.5 W/cm²) for 1 min but it showed the laser power of 0.2 W/cm² is used for the following studies on line 220. It should provide the temperature increase after applied with 0.2 W/cm² at FSCNO-DH nanoparticles (x µg/ml)
6. In this study, the authors used the same process to fabricate the FSCNT and FSGO nanoparticles with CNT and Graphene loaded, respectively and further demonstrated the FSCNO has a 2-3 times higher photothermal property than that of FSCNT and FSGO when the concentration is more than

0.25 mg/ml. However, SEM images of FSCNT nanoparticles and FSGO nanoparticles in Supplementary Figure 4. displayed their morphology much different from FSCNO, so it should provide any evidence that both FSCNT FSGO nanoparticles were formed. More importantly, in this study, the paper emphasize they used a low-NIR power to trigger drug release from the nanoparticle. What is the purpose? If they hope to use a lower power to raise temperature but not too high, it is pretty suitable to use FSCNT and FSGO. Please explain it.

7. In Figure 4b and 4c, it showed that compared to that without laser radiation, the DOX fluorescence in nuclei is highly increased after NIR irradiation of cells treated with the FSCNO-DH nanoparticles. Please illustrated the possible reaction and mechanism why the laser can enhance DOX release ?

8. In contrast to Question-6, On line 218, The paper mentioned that the superior photothermal effect of FSCNO nanoparticles could effectively enhance the retention of DOX in the drug resistant cells. Please illustrate why the retention of DOX can be enhanced using the NIR photothermal effect.

9. To confirm that fucoidan can target the P-selectin on the tumor, the authors activated the human umbilical vein endothelial cells (HUVECs) to mimic the tumor vasculature and demonstrated that the expression of P-selectin is evident in activated HUVECs (aHUVECs) in Fig. 5a. However, the P-selectin does not only appeared on the tumor but also widely expressed on the vasculature in other organs such as heart. Therefore, when the FSCNO was subjected to Intravenous injection through the vessel, how the FSCNO can precisely delivered and targeted to tumor?

10. Finally, in Figure 7, it showed that FSCNO-DH+L with co-injection of aHUVECs (With EC showing augmented antitumor efficacy in mice, which is attributed to due to the capability of the FSCNO-DH nanoparticles in targeting tumor vasculature and enhancing the targeted delivery of HM to inhibit the efflux pump in drug resistant tumor when combined with the NIR laser irradiation. However, the effect does not appear in the A2780ADR cells. In addition, in real clinic application, they will inject the FSCNO-DH nanoparticles to treat the orthotopic tumors without using co-injection of aHUVECs. So if possible, please give comments if the FSCNO-DH nanoparticles can still obtain a better therapeutic efficacy in orthotopic tumors of mice.

Line 83, typing error, silica (S) surface-decorated with Fucoidan. It should be Si instead of S.

Line 332, typing error, it should be “ conclusion” instead of “Discussion”

Point-by-point response to reviewers

We would like to thank all the reviewers for their insightful and thoughtful comments! We have revised the manuscript according to their advices, which should significantly improve the clarity and quality of our work. Below is a list of the point-by-point responses to the reviewer comments and the corresponding changes that we made. All the changes are highlighted in the manuscript, as well.

Reviewer #1: expert in nanoparticles

The paper reports the construct of a fucoidan-decorated silica-carbon nano-onion (FSCNO) hybrid nanoparticle for targeting tumor vasculature and P-glycoprotein to overcome cancer drug resistance. Different in vitro and in vivo models have been used to investigate the targeting efficacy, very interesting and promising results were obtained. However, several key questions need to be addressed:

Re: We thank the reviewer for the insightful and thoughtful comments!

1. Fucoidan is adsorbed on the nanoparticle surface through electrostatic interactions. How stable are this surface coating in vitro and in vivo? As it is known that electrostatic interaction is not strong enough to ensure the stability of surface coating, when in salt or cell culture medium, the surface coating could falls. Also, what's the density of Fucoidan on the particle surface? Does the Fucoidan density on the surface affect their tumor vasculature targeting?

Re: We understand the concern and conducted more experiments to address it! To confirm the stability of the fucoidan coating on the FSCNO-DH nanoparticles, they were incubated in cell culture medium for 2 days before investigating their capability of targeting the aHUVCEs versus HUVECs under both static culture in Petric dish and dynamic culture in the microfluidic device. As show in the new Supplementary Fig. 19a for static culture or 19 b for dynamic culture of aHUVCEs, the FSCNO-DH nanoparticles after incubating with cell culture medium can still target the aHUVCEs regardless of the culturing conditions. This is probably due to the strong interaction between the sulfonyl group (in fucoidan) and amino group (on ASCNO nanoparticles) by reducing the nucleophilicity and basicity of the amino group to form sulfonamide-like stable interaction, which has been widely used to protect amino group from other groups (such as carbonyl group, Refs. 36-37). Since the overall negatively charged materials in blood or cell culture medium (e.g., proteins or nuclei acids) are mainly formed with carbonyl group, the FSCNO nanoparticles are stable in them.

We agree that the density of fucoidan may affect the targeting capability of the FSCNO nanoparticles. In this study, fucoidan was added to the ASCNO nanoparticles until the zeta potential of FSCNO nanoparticles was stable (-38.0 ± 4.1 mV), suggesting the fucoidan have probably fully decorated the surface of the nanoparticles. In order to show the fucoidan density is important for the targeting effect, FSCNO nanoparticles with reduced/low fucoidan (F^{low} SCNO nanoparticles) were prepared. As shown in the new Supplementary Fig. 20a, the zeta potential of the F^{low} SCNO-DH nanoparticles is increased to -11.2 ± 3.8 mV if the amount of fucoidan during the nanoparticle preparation is reduced from 5 mg to 2 mg. Consequently, the targeting capability of the F^{low} SCNO-DH nanoparticles is weakened compared with the FSCNO-DH nanoparticles (the new Supplementary Fig. 20b versus Figure 5b or Supplementary Fig. 19a). The aforementioned discussions are now incorporated in lines 4-15 on page 12 of this revision and the results are shown in new Supplementary Figure 19 and 20.

2. Drug loading: as the FSCNO nanoparticles have a CNO core and silica shell coated with Fucoidan. What's the mechanism of drug (DOX and HM) loading? Where are the drugs loaded? On the particle surface? DOX, a water soluble drug, was firstly loaded, what's the interaction between DOX and the

particle? Does the DOX coating cover the Fucoidan? Also, following the loading of DOX, then HM in DMSO was further mixed with the nanoparticles, is the HM also on the particle surface covering the DOX? as HM is hydrophobic, what forces or interactions keep the hydrophobic HM on the particle surface then? What's the drug loading (the ratio of drug to the drug-loaded nanoparticle mass)? The layer DOX and HM should have covered the Fucoidan, how these coatings affect the targeting effect of Fucoidan?

Re: Sorry for the confusion! DOX and HM should not locate on the surface of the nanoparticles as we always washed the nanoparticles at least three times with DI water after drug encapsulation. The reverse microemulsion method was used in this study to synthesize the SCNO nanoparticles. The silica nanoparticles prepared with this method has been used to encapsulate both hydrophilic and hydrophobic drugs (Refs. 39-40). To confirm the FSCNO nanoparticles have spaces for encapsulating drugs, nitrogen gas (N₂) sorption measurements were carried out to investigate both the surface area and pore size distribution within the nanoparticles. The N₂ adsorption isotherm of the FSCNO nanoparticles (shown in the new Supplementary Fig. 6a) is typical of the type IV isotherm that indicates the micro- or mesoporous feature according to the International Union of Pure and Applied Chemistry (IUPAC) classification (Ref. 31). The surface areas within the FSCNO nanoparticles are 250.3 m² g⁻¹ and the average pore size is 3.2 nm with two peaks at ~1.4 nm and ~3.1 nm (see the new Supplementary Fig. 6b). This porous structure within the FSCNO nanoparticles renders their capability of drug loading.

The loading of drugs inside the nanoparticles rather than on their external surface is further supported by the new zeta potential data. As shown in the new Supplementary Fig. 7a, the zeta potential of the two drugs-laden FSCNO-DH nanoparticles is similar to that of the FSCNO nanoparticles with no drug (Supplementary Fig. 5c), suggesting the drugs are not coated on the surface of the nanoparticles.

The encapsulation of DOX is probably because of the electrostatic interaction between DOX (positively charged) and silica (negatively charged) and the π - π stacking interaction between DOX and CNO. Although the exact mechanisms for the encapsulation of hydrophobic agents like HM in silica is still not well understood (Ref. 41), one possible mechanism is the electrostatic interaction between the negatively charged silica and the local positive charge (e.g., the amino group) in the HM molecule. Another possible mechanism is the π - π stacking between the naphthalene-like structure in HM and the tetracene structure in DOX and CNO. Therefore, both DOX and HM should be encapsulated inside the FSCNO nanoparticles and did not cover on the fucoidan, with a loading content of 3.3 ± 0.1 for HM and 4.3 ± 0.1 for DOX at a feeding ratio of 1:20 (HM or DOX : nanoparticles). The aforementioned discussions are now incorporated in first paragraph on page 6, lines 1-2, 6-10 in first paragraph of Discussion section, and first 2 lines on page 17 of this revision and the results are shown in new Supplementary Figure 6 and 7a.

3. Page 9, line 219, "This is probably due to the superior photothermal effect of FSCNO nanoparticles, which could effectively trigger the release of HM to inhibit the P-gp and enhance the retention of DOX in the drug resistant cells for the observed effective destruction of the cells" How the NIR irradiation affect drug release? What's the trigger release mechanism? Were both the drugs capable of trigger release? Does the laser power affect the integrity of the nanoparticles?

Re: As per the reviewer's advice, we conducted the drug release profile of FSCNO-DH nanoparticles without laser irradiation. As shown in the new Figure 3e, the release of HM and DOX from the nanoparticles is minimal at least for 10 h while a triggered release can be precisely controlled with NIR irradiation. The possible mechanism for the NIR laser-controlled drug release is attributed to the laser irradiation induced heating. We have shown that the triggered drug releases from silica-fullerene (note: fullerene is a carbon nanomaterial as with nano-onion) hybrid nanoparticles is mainly because of the temperature increase in a previous study (Ref. 38). Similarly, the temperature in the FSCNO-DH

nanoparticles will increase during the laser irradiation, which should cause the release of the drugs out of the nanoparticles. The laser irradiation does not affect the integrity of the nanoparticles according to the TEM image of FSCNO nanoparticles after laser irradiation (see the new Supplementary Figure 10). The aforementioned discussions are now incorporated in lines 10-12 on page 8 and lines 2-7 on page 17 and the results are shown in new Figure 3e and new Supplementary Figure 10.

4. For in vivo targeting, the surface property of the nanoparticles plays a critical role. What's the surface property of the nanoparticles after drug loading? Are they a mixture of DOX, HM and Fucoidan? How are they distributed on the nanoparticle surface? and what are their densities on the particle surface? HM is very hydrophobic, which could affect the circulation of nanoparticles adversely? As hydrophobic nanoparticles could be quickly cleared from the circulation once injected.

Re: We agree the surface property of nanoparticles play a critical role for in vivo targeting! Per the reviewer's advices, we conducted more experiments to characterize the surface property of our nanoparticles, as detailed in our responses to Comment 2. Briefly, the DOX and HM should be inside the nanoparticles, therefore, encapsulation of the two drugs should have minimal impact on the targeting capability of the FSCNO nanoparticles. This is supported by the similar zeta potential of FSCNO and FSCNO-DH nanoparticles (see the new Supplementary Figure 7a versus Supplementary Figure 5c). The targeting capability of fucoidan on FSCNO-DH nanoparticles is confirmed by the data on in vivo imaging and antitumor capability. As shown in Fig. 6e-g and Supplementary Fig. 22, the FSCNO-I nanoparticles can target the tumor blood vessels formed from both the host endothelial cells and injected aHUVCEs. Furthermore, the FSCNO-DH+L treatment is more effective in killing tumors co-injected with aHUVCEs, confirming the function of fucoidan decorated on the surface of FSCNO-DH nanoparticles. The aforementioned discussions are now incorporated in lines 10-13 on page 6 of this revision and the results are shown in new Supplementary Figure 7a.

5. In in vivo experiments, the authors used indocyanine green loaded nanoparticles for investigating tumor targeting capability? How was ICG loaded? Are they also loaded on the particle surface through absorption? Then the surface property of ICG loaded nanoparticles should be very different from DOX and HM loaded nanoparticles, as ICG has very different property from HM.

Re: Sorry for the confusion! Similar to DOX, ICG was mixed with FSCNO nanoparticles at a feeding ratio of 1:20 (ICG : nanoparticles in weight) in deionized water for 24 h and then washed with deionized water for three times to remove non-encapsulated ICG. Although the property of ICG is quite different with HM, the zeta potential of ICG-laden FSCNO nanoparticles (FSCNO-I) is similar to that of FSCNO or FSCNO-DH nanoparticles as shown in new Supplementary Figure 7b, suggesting the therapeutic agents are not on the surface of the nanoparticles. The aforementioned discussions are now incorporated in lines 3-4 in the 2nd paragraph on page 12, and 2nd paragraph on page 20 of this revision and the results are shown in new Supplementary Figure 7b.

6. For in vivo antitumor capacity experiments, it can be seen that FSCNO-DH had very limited effect in inhibiting the tumor growth for all the three cells, so it would be very helpful to have a control experiment of FSCNO + laser to demonstrate that just targeted photothermal treatment is sufficient.

Re: In this study, the photothermal treatment is designed to trigger the drug release at tumor site but not kill the tumor cells with heat. Therefore, we have extensively investigated the photothermal effects of FSCNO nanoparticles to NCI/ADR-RES, A2780ADR and OVCAR-8 cells at various nanoparticle concentrations and laser powers. As shown in the new Supplementary Figure 16a-b, only the highest power (1 W cm^{-2}) combined with empty FSCNO nanoparticles could effectively kill all the three different types of cells (Supplementary Fig. 16a). Mild photothermal effect (0.5 , 0.2 , and 0.05 W cm^{-2}) has minimal

impact on the viability of tumor cells (Supplementary Fig. 16a-c). Since we only applied a low laser power of 0.2 W cm^{-2} for the in vivo study, the empty FSCNO nanoparticles combined with laser irradiation should have minimal effects on the tumor growth. The aforementioned discussions are now incorporated in lines 10-12 on page 14 of this revision and the results are shown in new Supplementary Figure 16a-b. Again, we thank the reviewer for all the insightful and thoughtful comments!

Reviewer #2: expert in nano-onions

The manuscript of Wang et al. reports on the development of fucoidan-decorated silica and CNO hybrid (FSCNO) nanoparticle and their potential application for selective targeting of tumor vasculature and anticancer effect, thanks to the co-deliver of a P-gp inhibitor and a chemotherapeutic drug.

This is a very interesting paper, which clearly shows the potential biomedical applications of carbon-based nanomaterials. The studies prove the selective targeting of tumor vasculature on three different model: a cell culture, a microfluidic dynamic culture and finally in vivo on a tumoral mice model.

The delivery of P-gp inhibitors directly to multidrug-resistant cancer cells is a clever idea that has allowed overcoming drug resistance both in vitro and in vivo. Moreover, the authors demonstrated the capability of these new smart nanohybrids to release the anticancer drug upon internal trigger into the tumor area, by using the high light absorption of carbon nano-onions in the NIR. The manuscript is well written, the subject is new and the data are solid.

The scope of the problem tackled by the paper significant enough to warrant a Nature publication. I believe that this paper matches the quality criteria requested for publication in Nature Communications and therefore suggest its publication after minor corrections.

Re: We thank the reviewer for the insightful and thoughtful comments!

1. While it is a clever idea to coat the CNOs with TEOS and APTMS to improve their solubility and loading capacity and the Silica-CNO nanoparticles (SCNO) produced look uniform in the TEM images (Supp. Fig. 2), the TEM images of the starting material shown in Figure 2b are not convincing of CNO. The author comment “the nano-onion structure (~50 nm in diameter) can be observed only at the edge of the aggregates at higher magnification (right, Fig. 2b). A better HRTEM image of individual CNO should replace Fig. 2b. It is also a little confusing the following statement “Interestingly, the hybrid SCNO nanoparticles are smaller than the CNOs”. The authors are invited to carefully revise.

Re: As per the advice, we conducted more experiments and added HRTEM images of CNO as the new Supplementary Figure 2 to better show the multilayered onion-like structure of CNO.

We also conducted more experiments to under why the size of the FSNO nanoparticles is smaller than CNO nanoparticles. According to the Fourier-transform infrared spectroscopy (FTIR) spectrum shown in the new Supplementary Fig. 4a, the formation of the Si-O-C bond at 954 and 1070 cm^{-1} is observable for the FSNO nanoparticles. This indicates the reaction between TEOS and CNO, which may make the CNO structure more compact. The aforementioned discussions are now incorporated in last line on page 4 and lines 1, 3-5 on page 5 of this revision and the results are shown in new Supplementary Figure 2 and 4.

2. There are typos and errors in the paper, the reviewer recommends proofreading the paper an additional time missing unit on line 129 & 130. Is line 287 is supposed to say ' Similar to in vitro imaging', rather than 'Similar to in vivo imaging' ? Error on line 311, etc.

Re: All the typos mentioned by the review are corrected now. Per the reviewer's advice, we also carefully proofread the entire manuscript to minimize typos and errors.

3. The DLS (Fig. 2d) should be given as a line plot instead of a scatter plot, to better see the curve.

Re: We agree and the DLS data are given in line plot now.

4. The UV-Vis solvent is not given.

Re: Sorry for the overlook! Per the advice, the UV-Vis solvents for the various groups are now given in the caption of Figure 2. Again, we thank the reviewer for all the insightful and thoughtful comments!

Reviewer #3: expert in p-selectin targeted nanoparticles

In this article, the authors reported a fucoidan-decorated silica-carbon nano-onion (FSCNO) hybrid nanoparticle that co-deliver P-gp inhibitor and chemotherapeutic drug (DOX) for precisely targeting tumor vasculature by specifically releasing P-gp inhibitor and anticancer drug into tumor cells. The carbon nano-onion was used to trigger the drug release via NIR and Silica surface-decorated with fucoidan (F) was used as a target to specifically bind to P-selectin that is overexpressed on tumor vasculature in this nanopatform (FSCNO). DOX is a chemotherapeutic drug.

The most contribution in this paper is that the authors try to achieve an effective inhibition on P-gp vis using P-gp inhibitor to reduce multi-drug resistant. The functions of the components used for the synthesis of the nanoparticles are well known and the paper indeed made a well detailed in-vitro and in-vivo experiments to prove the FSCNO nanoparticle with the above-mentioned functions. The main claim in this paper is emphasized to control the release of P-gp inhibitor via NIR for effectively inhibiting the P-gp. However, what is the novelty I am thinking? Combination of fucoidan and P-gp release to exhibit the targeting of tumor vasculature and P-gp inhibition indeed contains somewhat important contribution and novelty but is not so highly obvious enough. Some questions and concerns about the paper are listed below.

Re: We thank the reviewer for the critical comments! Besides the targeted inhibition of P-gp in tumor to significantly improve chemotherapy against cancer drug resistance, this work is novel in terms of the materials (HM and nano-onion) used, the discovery of the superior photothermal property of the nano-onion, the three different models for characterizing the targeting of tumor vasculature, and the method of establishing in vivo tumor models with different levels of P-selectin expression, as detailed below.

1) HM30181A (HM) is a third-generation inhibitor of P-gp. However, HM is insoluble in water and was used only for oral delivery to inhibit P-gp in the intestinal endothelium in previous studies. It has never been used to inhibit the P-gp in multidrug resistant cancer cells.

2) Due to the highly hydrophobic nature, CNOs have poor solubility in water or even commonly used organic solvents (e.g., benzene and toluene) and they tend to form large aggregates in these solvents. This greatly limits the biomedical applications of CNOs. In this study, we resolved this problem by forming CNO-silica hybrid nanoparticles. Furthermore, we discovered the superior photothermal effect of CNOs in aqueous solution compared with carbon nanotube (CNT) and graphene oxide (GO) that have been widely studied for various biomedical applications including photothermal therapy.

3) Three different models were used to study the tumor vasculature targeting effect of our nanoparticles: static culture of activated (in tumor) versus non-activated (in normal tissue) HUVECs in Petri dish,

dynamic culture of activated versus non-activated human endothelial cells in microfluidic device, and three different kinds of human tumors in mice. In particular, the microfluidic model developed in this study may be valuable for mimicking the in vivo perfusion in blood vessels to examine the targeting capability of nanoparticles.

4) The activated HUVECs (aHUVECs) were co-injected with three different kinds of tumor cells to mimic the tumors with different intensity of P-selectin for the in vivo studies. Both the in vivo imaging and the therapeutic results further confirmed the targeting capability of fucoidan to P-selectin on tumor blood vessels. Therefore, our study should be an important reference for the future P-selectin mediated drug delivery.

The aforementioned clarifications are given on lines 2-5 in the 2nd paragraph on page 4, lines 2-4 in the 2nd paragraph on page 5, third paragraph on page 17, and lines 7-8 on page 18. Sorry for the confusion! With these clarifications, we sincerely hope that the reviewer agrees that the novelty of this work is high enough and appropriate for publication in *Nature Communications*.

1. The particle size of FSCNO is very small about 20-25 nm and Transmission electron microscopy (TEM) images of the CNOs in Figure 2 (b) and FSCNO nanoparticles (c) showed its morphology looks like a solid particle (not hollow structure) with a very small capacity to load both Pgp inhibitor and chemotherapeutic drug (DOX) inside the nanoparticle. So Please provide the loading capacity and nature release of both drugs

Re: To address the concern on the internal structure of the FSCNO nanoparticles for encapsulating drugs, nitrogen gas (N₂) sorption measurements were carried out to investigate both the surface area and pore size distribution within the nanoparticles. The N₂ adsorption isotherm of the FSCNO nanoparticles (shown in the new Supplementary Fig. 6a) is typical of the type IV isotherm that indicates the micro- or mesoporous feature according to the International Union of Pure and Applied Chemistry (IUPAC) classification (Ref 31). The surface areas within the FSCNO nanoparticles are 250.3 m² g⁻¹ and the average pore size is 3.2 nm with two peaks at ~1.4 nm and ~3.1 nm (new Supplementary Fig. 6b). This porous structure within the FSCNO nanoparticles renders their capability of drug loading The silica nanoparticles prepared with this method has been used to encapsulate both hydrophilic and hydrophobic drugs. (Refs 39-40). Per the reviewer's advice, we have added the loading capacity (3.31 ± 0.08 for HM and 4.26 ± 0.07 for DOX at a feeding ratio of 1:20, HM or DOX : nanoparticles) and natural release of both drugs. As shown in the new Figure 3e, the natural release of both drugs from FSCNO-DH nanoparticles is minimal at least for 10 hours.

The aforementioned discussions are now incorporated in lines 1-9 on page 6, line 12 on page 8, and first two lines in first paragraph of Discussion section of this revision and the results are shown in new Figure 3e, Supplementary Figure 6 and 7a.

2. Another question is how both hydrophobic HM and doxorubicin hydrochloride (??) were encapsulated in the hydrophobic CNO since HM is hydrophobic and doxorubicin hydrochloride tend to be dissolved in hydrophilic solvent like water so both drugs are very difficult physicochemical characters. According the experimental illustration in this paper, they mentioned that DOX first mixed with FSCNO nanoparticles at a feeding ratio of 1:20 (DOX : nanoparticles in weight) in deionized water for 24 h. Then, the nanoparticles were collected and further mixed with HM dissolved in DMSO (1 mg/ml) (HM : nanoparticles) in deionized water for 24 h. Therefore, I think that the authors should provide more explanation to illustrate how both drugs were encapsulated and loading capacity of HM and Dox.

Re: We understand and provide more explanations per the advice! Mesoporous silica nanoparticles have been used to encapsulate various hydrophilic and hydrophobic drugs in the literature (Refs. 39-40). In this

study, the sequence of encapsulating DOX and HM in the FSCNO nanoparticles was optimized. The encapsulation efficiency of HM is only ~24% if mixing HM with the FSCNO nanoparticles first, which is much lower than that (~69%) of the optimized method of mixing DOX with the nanoparticle first. This suggests that DOX might have some interactions with HM to enhance the encapsulation of HM inside the FSCNO nanoparticles. The encapsulation of DOX is probably because of the electrostatic interaction between DOX (positively charged) and silica (negatively charged) and the π - π stacking interaction between DOX and CNO. Although the exact mechanisms for the encapsulation of HM in silica is still not well understood (Ref. 41), one possible mechanism is the electrostatic interaction between the negatively charged silica and the local positive charge (e.g., the amino group) in the HM molecule. Another possible mechanism is the π - π stacking between the naphthalene-like structure in HM and the tetracene structure in DOX and CNO. The aforementioned discussions are now incorporated in first 12 lines in Discussion section of this revision.

3. Next, the control release for both Pgp inhibitor and chemotherapeutic drug (DOX) from the FSCNO nanoparticle is very important. Figure 3(e) only showed both DOX and HM are simultaneously released only under the trigger with near infrared (NIR) laser irradiation (0.5 W/cm²) for 1 min. However, how much FSCNO nanoparticles can be uptaken by the cancer cells since the FSCNO nanoparticles have a very high negative-charged surface, which is usually not favorable for cell uptake. They should provide the data measured by Flow cytometry

Re: As per the reviewer's advice, we have investigated the cellular uptake of the FSCNO-DH nanoparticles using flow cytometry. As shown in the new Supplementary Figure 14a-b, the DOX in NCI/RES-ADR and A2780ADR cells is minimal if the cells were treated with free DOX and greatly increased when free DOX is combined with HM (DOX+HM) for treating the cells. The DOX fluorescence intensity in FSCNO-DH+L treated cells is stronger than that in FSCNO-DH treated cells, probably due to the self-quenching effect of DOX fluorescence in the FSCNO nanoparticles (Fig. 2g). Similar to the microscopy data, HM has minimal effect on the DOX uptake of OVCAR-8 cells (see the new Supplementary Fig. 14c). Overall, these results suggest that the FSCNO-DH nanoparticles could be efficiently taken up by both non-drug resistant and multidrug resistant tumor cells and deliver drugs into them. The aforementioned discussions are now incorporated in last 7 lines in the first paragraph on page 9.

4. Another more important question is that as the FSCNO was uptaken by the cancer cells, how the release of P-gp inhibitor and chemotherapeutic drug from FSCON nanoparticles can be effectively controlled ?? Here, it only showed both drugs are released simultaneously in Figure 3(e). However, it is better to release HM first to inhibit the P-gp and then the released DOX can be avoided to be pumped out. Please explained it in detail. Otherwise, The FSCNO nanoparticles can not achieve direct inhibition of the drug efflux pumps of DOX for overcoming the multidrug resistance of cancer.

Re: When controlling the drug release in this study, HM and DOX are triggered to release simultaneously from the FSCNO nanoparticles during the NIR irradiation. Nonetheless, HM can quickly, selectively, and potently inhibit the P-gp function. As a result, although some DOX may be pumped out of cells before the P-gp function is inhibited, it could partially re-enter the cells after the HM binds with the P-gp. As shown in Figure 3a, the multidrug resistant capability of NCI/RES-ADR and A2780ADR cells was inhibited when they are treated with the mixture of DOX and HM. Ideally, it is better to release HM first to inhibit the P-gp function and then release DOX to better keep the DOX inside cells. We thank the reviewer for this advice and future studies on this capability is warranted. The aforementioned discussions are now incorporated in 2nd paragraph of Discussion section on page 17.

5. Figure 3(e) showed the drug release triggered laser at 0.5 W/cm²) for 1 min but it showed the laser

power of 0.2 W/cm² is used for the following studies on line 220. It should provide the temperature increase after applied with 0.2 W/cm² at FSCNO-DH nanoparticles (x µg/ml).

Re: As per the reviewer's advice, we conducted more experiments to quantify the temperature changes after applying the NIR laser power of 0.2 W/cm² and the data are shown in new Supplementary Figure 17. The aforementioned discussions are now incorporated in last three lines in the first paragraph on page 10.

6. In this study, the authors used the same process to fabricate the FSCNT and FSGO nanoparticles with CNT and Graphene loaded, respectively and further demonstrated the FSCNO has a 2-3 times higher photothermal property than that of FSCNT and FSGO when the concentration is more than 0.25 mg/ml. However, SEM images of FSCNT nanoparticles and FSGO nanoparticles in Supplementary Figure 4. displayed their morphology much different from FSCNO, so it should provide any evidence that both FSCNT FSGO nanoparticles were formed. More importantly, in this study, the paper emphasize they used a low-NIR power to trigger drug release from the nanoparticle. What is the purpose? If they hope to use a lower power to raise temperature but not too high, it is pretty suitable to use FSCNT and FSGO. Please explain it.

Re: As per the reviewer's advice, we conducted more experiments to confirm the successful synthesis of SCNT and SGO nanoparticles. As shown in the new Supplementary Figure 4, the FTIR spectra of the SCNO, SCNT or SGO nanoparticles exhibit similar absorption bands corresponding to Si-O-Si (ν_s at 800 cm⁻¹ and ν_{as} at 1080-1200 cm⁻¹) and Si-O-C (ν_s at 954 cm⁻¹ and ν_{as} at 1070 cm⁻¹), suggesting the formation of silica structure and the reaction between TEOS with CNO, CNT, or GO. This is further confirmed with the SEM image which shows the tube structure of CNT is changed after reacted with silica and coated with fucoidan to form the FSCNT nanoparticles. Although the sheet structure of GO is difficult to identify in the SEM image because they are very thin, the thickness of the sheet structure is significantly increased and more visible after reacting with silica and coating with fucoidan to form the FSGO nanoparticles (Supplementary Figure 8). These data suggest the successful formation of FSCNO and FSGO nanoparticles.

Due to the superior photothermal effect of FCNOs, we could use a lower power to achieve enough heat to trigger the drug release. However, people may have to use a higher NIR laser power to trigger the drug release if used less efficient photothermal nanomaterials (such as FSCNT or FSGO). It is worth noting that the NIR laser irradiation alone could increase the temperature of tissues. Due to the superior photothermal effect of FCNO nanoparticles, an NIR laser power as low as 0.2 W cm⁻² was used for the in vivo studies in this work, which could avoid the potential damage to normal tissue (with no nanoparticles) at a higher power (e.g., 1 W cm⁻², new Supplementary Fig. 25). The aforementioned discussions are now incorporated in lines 15-19 on page 7 and lines 8-12 on page 16 and the results are shown in new Supplementary Figure 4 and 25.

7. In Figure 4b and 4c, it showed that compared to that without laser radiation, the DOX fluorescence in nuclei is highly increased after NIR irradiation of cells treated with the FSCNO-DH nanoparticles. Please illustrated the possible reaction and mechanism why the laser can enhance DOX release?

Re: There are two reasons that may contribute to the highly increased DOX in nuclei of cells treated with FSCNO-DH after NIR laser irradiation. First, NIR laser irradiation can trigger the drug release. We have found that the triggered drug releases from silica-fullerene (note: fullerene is a carbon nanomaterial as with nano-onion) hybrid nanoparticles is mainly because of the temperature increase in a previous study (Ref. 38). Similarly, the temperature in the FSCNO-DH nanoparticles will increase during the laser irradiation, which should cause the release of the drugs out of the nanoparticles. Second, the released HM after laser irradiation could inhibit the pumping capability of P-gp and keep the DOX inside the multidrug

resistant cells. As shown in Figure 4b, the DOX fluorescence in nuclei of FSCNO-D treated cells after NIR irradiation is weaker than FSCNO-DH treated cells (Figure 4c), suggesting the inhibition of the P-gp function with HM is important to improve the bioavailability of DOX in cells. The aforementioned discussions are now incorporated in lines 2-7 on page 17.

8. In contrast to Question-6, On line 218, The paper mentioned that the superior photothermal effect of FSCNO nanoparticles could effectively enhance the retention of DOX in the drug resistant cells. Please illustrate why the retention of DOX can be enhanced using the NIR photothermal effect.

Re: We are sorry for the confusion! The retention of DOX is not directly enhanced by the photothermal effect, but due to the triggered release of HM from the FSCNO-DH nanoparticles after laser irradiation. The superior photothermal effect of FSCNO nanoparticles could effectively trigger the release of HM to inhibit the P-gp and then enhance the retention of DOX in the multidrug resistant cancer cells. This is further confirmed with flow cytometry data. The DOX in NCI/RES-ADR and A2780ADR cells is minimal if they are treated with free DOX alone and greatly increases when they are treated with FSCNO-DH. Importantly, the fluorescence intensity in FSCNO-DH+L treated cell is stronger than that in FSCNO-DH treated cells. The aforementioned discussions are now incorporated in last 7 lines in the first paragraph on page 9 and last 6 lines in first paragraph on page 10.

9. To confirm that fucoidan can target the P-selectin on the tumor, the authors activated the human umbilical vein endothelial cells (HUVECs) to mimic the tumor vasculature and demonstrated that the expression of P-selectin is evident in activated HUVECs (aHUVECs) in Fig. 5a. However, the P-selectin does not only appeared on the tumor but also widely expressed on the vasculature in other organs such as heart. Therefore, when the FSCNO was subjected to Intravenous injection through the vessel, how the FSCNO can precisely delivered and targeted to tumor?.

Re: We agree the FSCNO nanoparticles may bind with the P-selectin expressed in other normal tissues and not all the nanoparticles can accumulate at the tumor site. In fact, it is still difficult to find the exclusive maker that exists only in tumor cells or tumor microenvironment currently. As shown in Figure 6c-d for the in vivo imaging, FSCNO nanoparticles could deliver more imaging agent to tumors if co-injecting aHUVECs with tumor cells for grow in vivo tumors, suggesting the FSCNO nanoparticle can target P-selectin for tumor accumulation. Another mechanism is the Enhanced Permeability and Retention (EPR) effect of tumor vasculature compared to the vasculature in normal tissues/organ, which has been widely used for developing nanomedicine to passively target tumor over normal tissues/organs.

It is worth noting that some nanoparticles are distributed in liver, kidney and probably other organs according to Figure 6b-c. However, there is no obvious side effects observed in the FSCNO-DH+L treated mice (Supplementary Figure 24). This may be due to the NIR irradiation-controlled drug release inside tumors and minimal drug release outside tumors with no NIR irradiation (Fig. 3e). The aforementioned discussions are now incorporated in lines 2-3 in the last paragraph on page 15.

10. Finally, in Figure 7, it showed that FSCNO-DH+L with co-injection of aHUVECs (With EC showing augmented antitumor efficacy in mice, which is attributed to due to the capability of the FSCNO-DH nanoparticles in targeting tumor vasculature and enhancing the targeted delivery of HM to inhibit the efflux pump in drug resistant tumor when combined with the NIR laser irradiation. However, the effect does not appear in the A2780ADR cells. In addition, in real clinic application, they will inject the FSCNO-DH nanoparticles to treat the orthotopic tumors without using co-injection of aHUVECs. So if possible, please give comments if the FSCNO-DH nanoparticles can still obtain a better therapeutic efficacy in orthotopic tumors of mice.

Re: We are sorry for the confusion! There is no significant difference between the A2780ADR cells With

EC and W/O EC groups for the FSCNO-DH+L treatment. This is mainly because the A2780ADR tumors in With EC group grow faster than that in the W/O EC group. In other words, the decrease in tumor volume for the With EC group after the FSCNO-DH+L treatment is much more than that for the W/O EC group (Supplementary Fig. 23b), demonstrating the importance of tumor vasculature targeting for enhancing the efficacy of nanoparticle-mediated cancer therapy.

The co-injection of aHUVeCs with tumors cells was to mimic the condition of tumors with various degree of P-selectin expression. This is also helpful to confirm the importance of the P-selectin targeting capability of FSCNO nanoparticles for enhancing cancer therapy. Importantly, FSCNO-DH+L treatment can still significantly inhibit the growth of the NCI/RES-ADR, A2780ADR, and OVCAR-8 tumors without co-injection of the aHUVeCs, compared with the treatments of free drug, FSCNO-D+L, and FSCNO-DH. In other words, the FSCNO-DH nanoparticles with laser irradiation can efficiently inhibit the growth of multidrug resistant tumors in general and is even more effective for tumors with high P-selectin expression or ample blood vessels. The aforementioned discussions are now incorporated in last line on page 14, first paragraph on page 15, and third paragraph in the Discussion section according to the reviewer's advice.

Line 83, typing error, silica (S) surface-decorated with Fucoidan. It should be Si instead of S.

Re: We thank the reviewer for capturing the typo! It is now corrected and we also carefully proofread the entire manuscript to minimize typos/errors.

Line 332, typing error, it should be “ conclusion” instead of “Discussion”

Re: We agree the last section is more like a Conclusion than Discussion section! According to the journal format, the last section headline is called Discussion and there is no a Conclusion section. In this revision, we have added more discussions in this section so that it is more like a Discussion section. Again, we thank the reviewer for all the insightful and thoughtful comments!

REVIEWER COMMENTS

Reviewer #1 (Remarks to the Author):

The authors have addressed most of the questions I raised. I only have a minor point.

For the triggered drug release of HM and DOX, the authored explained that the triggered release was mainly due to the temperature increase as a result of the laser irradiation. On the other hands, the authors commented that the encapsulation of DOX and HM is because of electrostatic interaction, as well as possible pi-pi stacking. Does this mean electrostatic interaction or pi-pi stacking interaction would decrease with the increase of temperature? I am trying to make sense of the explanations. The mechanisms underpinning the drug encapsulation and controlled release are very critical for future design of useful nanomedicines.

Reviewer #2 (Remarks to the Author):

Dear Authors

I have gone through the revised manuscript. The reviewer's suggestions/ comments have been addressed; therefore in my opinion the revised manuscript may be accepted for publication in its present form.

Reviewer #3 (Remarks to the Author):

Although the authors have tried to respond the questions, some of the questions were not well addressed and need to be further illustrated as follows.

1. Although silica nanoparticles prepared with this method has been used to encapsulate hydrophilic or hydrophobic drugs, it is still very difficult to load both hydrophilic and hydrophobic drugs. The authors introduce two refs. to illustrate silica nanoparticles can be loaded with hydrophobic or hydrophilic drugs. However,

Ref. 39: Hui Y, et al. Role of Nanoparticle Mechanical Properties in Cancer Drug Delivery.. *Acs Nano* 13,7410-7424 (2019). It is a review paper and mentioned that different silica precursors was used for making soft and stiff SNCs and different surface groups can be introduced onto the surfaces of the SNCs.

Ref. 40: *ACS Appl Mater Interfaces* 12, 4308-4322 (2020).

A near-infrared (NIR)-responsive photosensitizer, indocyanine green (ICG), was loaded into the MSNs for the local generation of ROS to enhance cytosolic RNA delivery.

Both papers do not mention that silica can be loaded simultaneously both hydrophilic and hydrophobic drugs.

2. To respond how both drugs were encapsulated and loading capacity of HM and Dox.

This paper mentioned that they tried the sequence of encapsulating DOX and HM in the FSCNO nanoparticles to see which condition can load a higher HM but without any scientific evidence or explanation.

In addition, please why mixing DOX with FSCNO nanoparticles using a feeding ratio of 1:20 (DOX : nanoparticles in weight) in deionized water for 24 h.

The loading capacity(%) (3.31 ± 0.08 for HM and 4.26 ± 0.07 for DOX/HM at a feeding ratio of 1:20, is very low, indicating that both drugs are not easily loaded into the nanoparticles. Please explain the above concerns.

3. The authors replied that it is very difficult to release HM first to inhibit the P-gp function and then release DOX to better keep the DOX inside cells. In other words, the multidrug resistant capability on the cancer cells using the nanoparticle will be much reduced. I wonder if the authors have any new idea to overcome the questions and support the study. Otherwise, the contribution of this paper to emphasize the multidrug resistant become less significant.

4. The Fucodan-based nanoparticle can target P-selectin for tumor accumulation. In addition, the P-selectin does not only appear on the tumor but also widely expressed on the vasculature in other organs. In this response, the authors do not reply the question when the FSCNO was subjected to Intravenous injection through the vessel, how the FSCNO can precisely delivered and targeted to tumor?

Point-by-point response to reviewers

We would like to thank all the reviewers for their insightful and thoughtful comments! We have revised the manuscript according to their advices, which should significantly improve the clarity and quality of our work. Below is a list of the point-by-point responses to the reviewer comments and the corresponding changes that we made. All the changes are highlighted in the manuscript, as well.

Reviewer #1

The authors have addressed most of the questions I raised. I only have a minor point.

For the triggered drug release of HM and DOX, the authored explained that the triggered release was mainly due to the temperature increase as a result of the laser irradiation. On the other hands, the authors commented that the encapsulation of DOX and HM is because of electrostatic interaction, as well as possible pi-pi stacking. Does this mean electrostatic interaction or pi-pi stacking interaction would decrease with the increase of temperature? I am trying to make sense of the explanations. The mechanisms underpinning the drug encapsulation and controlled release are very critical for future design of useful nanomedicines.

Re: We thank the reviewer for the insightful and thoughtful comments! Yes, we agree the possible mechanism for the NIR laser-controlled drug release is attributed to the change in the stability of the electrostatic and π - π stacking interactions (between HM/DOX and the surface within the nanoparticles) with the change of temperature. The Brownian motion of all molecules increases with temperature, which may destabilize the electrostatic interaction and π - π stacking to free the HM and DOX that may further diffuse out of the nanoparticle under concentration gradient. The aforementioned discussions are now incorporated in last 4 lines in the second paragraph of Discussion section.

Reviewer #2

Dear Authors

I have gone through the revised manuscript. The reviewer's suggestions/comments have been addressed; therefore in my opinion the revised manuscript may be accepted for publication in its present form.

Re: We thank the reviewer for the kind comments!

Reviewer #3

Although the authors have tried to respond the questions, some of the questions were not well addressed and need to be further illustrated as follows.

1. Although silica nanoparticles prepared with this method has been used to encapsulate hydrophilic or hydrophobic drugs, it is still very difficult to load both hydrophilic and hydrophobic drugs. The authors introduce two refs. to illustrate silica nanoparticles can be loaded with hydrophobic or hydrophilic drugs. However, Ref. 39: Hui Y, et al. Role of Nanoparticle Mechanical Properties in Cancer Drug Delivery.. Acs Nano 13,7410-7424 (2019). It is a review paper and mentioned that different silica precursors was used for making soft and stiff SNCs and different surface groups can be introduced onto the surfaces of

the SNCs.

Ref. 40: ACS Appl Mater Interfaces 12, 4308-4322 (2020). A near-infrared (NIR)-responsive photosensitizer, indocyanine green (ICG), was loaded into the MSNs for the local generation of ROS to enhance cytosolic RNA delivery. Both papers do not mention that silica can be loaded simultaneously both hydrophilic and hydrophobic drugs.

Re: Sorry for the confusion! As per the reviewer's advice, we have replaced the Refs 39 and 40 with three new references (39-41), which reported the encapsulation of DOX (hydrophilic) together with different hydrophobic drugs (i.e., quercetin, curcumin, and pheophorbide a) simultaneously into silica-based nanoparticles.

In this work, the successful encapsulation of both DOX and HM in the FSCNO nanoparticles is supported by the UV-Vis absorbance spectra (Figure 2f), the *in vitro* cell and *in vivo* animal studies with two multidrug resistance cells (Figures 4 and 7). To further confirm this directly, we conducted new Fourier-transform infrared spectroscopy (FTIR) experiments in this 2nd revision. As shown in the new Supplementary Fig. 26, the aromatic bonds at 1582 cm⁻¹ can be observed only in the DOX, FSCNO-D, and FSCNO-DH groups, suggesting successful encapsulation of DOX in the nanoparticles. Similarly, methyl and carboxyl peaks at 1438 and 1407 cm⁻¹ are observed mainly in the HM, FSCNO-H, and FSCNO-DH groups, confirming the existence of HM in the nanoparticles. Interestingly, multiple peaks associated with DOX and HM disappear or reduce after they are encapsulated inside the nanoparticles (i.e., FSCNO-D, FSCNO-H and FSCNO-DH). This might be due to the π - π stacking between FSCNO nanoparticles and DOX/HM that decreases the molecule vibrations, because the disappeared or reduced peaks associated with DOX or HM are not evidently affected if they are simply mixed with the FSCNO nanoparticles for the measurements (i.e., FSCNO+D, FSCNO+H, or FSCNO+D+H). Moreover, the π - π stacking between FSCNO nanoparticles and DOX is also supported by the fluorescence spectra shown in Fig. 2g. This is consistent with the literature (Refs 43-44), showing the fluorescence intensity of DOX decreases after it is encapsulated in nanoparticles that have π -stacking interactions with DOX, compared with free DOX. Probably due to these interactions between DOX/HM and nanoparticles, both DOX and HM are successfully encapsulated in the FSCNO nanoparticles. The aforementioned discussions are now incorporated in last 17 lines in the first paragraph of the Discussion section.

2. To respond how both drugs were encapsulated and loading capacity of HM and Dox.

This paper mentioned that they tried the sequence of encapsulating DOX and HM in the FSCNO nanoparticles to see which condition can load a higher HM but without any scientific evidence or explanation. In addition, please why mixing DOX with FSCNO nanoparticles using a feeding ratio of 1:20 (DOX : nanoparticles in weight) in deionized water for 24 h. The loading capacity(%) (3.31 ± 0.08 for HM and 4.26 ± 0.07 for DOX/HM at a feeding ratio of 1:20, is very low, indicating that both drugs are not easily loaded into the nanoparticles. Please explain the above concerns.

Re: We understand and provide more explanations per the advice! The encapsulation efficiency of HM is only ~24% if mixing HM with the FSCNO nanoparticles first, which is much lower than that (~69%) of the optimized method of mixing DOX with the nanoparticle first. The encapsulation efficiencies are carefully measured using UV-Vis absorbance of the two agents. The loading of HM in FSCNO nanoparticles is probably because of the electrostatic interaction between the negatively charged silica and the local positive charge (e.g., the amino group) in the HM molecule. Another possible mechanism is the π - π stacking between the naphthalene-like structure in HM and the tetracene structure in DOX and CNO. This is further confirmed by the FTIR spectra shown in the new Supplementary Fig. 26. Multiple peaks of the DOX and HM FTIR spectra disappear or reduce after they are encapsulated inside the nanoparticles, which might be due to the π - π stacking interactions between the FSCNO nanoparticles and DOX/HM to decrease the molecule vibrations.

The feeding ratio from 1:10 to 1:20 (drug : nanoparticles) has been widely used in the literatures and also in our previous studies (Refs. 11, 45 and 46). The encapsulation efficiencies of HM and DOX are $68.5 \pm 1.7\%$ and $89.0 \pm 1.5\%$, respectively, at a feeding ratio of 1:20 (HM or DOX : nanoparticles), suggesting both drugs can be efficiently encapsulated in the nanoparticles. The total loading content for DOX and HM is $\sim 7.6\%$ that is also acceptable as the maximum loading content is only $\sim 9.1\%$ ($1/[1+10]$) with an encapsulation efficiency of 100% (the total feeding ratio is 1:10). The aforementioned discussions are now incorporated in the first 4 lines on page 6, last 13 lines in the first paragraph of Discussion section, and line 7 on page 21.

3. The authors replied that it is very difficult to release HM first to inhibit the P-gp function and then release DOX to better keep the DOX inside cells. In other words, the multidrug resistant capability on the cancer cells using the nanoparticle will be much reduced. I wonder if the authors have any new idea to overcome the questions and support the study. Otherwise, the contribution of this paper to emphasize the multidrug resistant become less significant.

Re: To address this comment, we conducted new experiments to investigate the cellular uptake and cytotoxicity of two multidrug resistance cells with different sequences of DOX and HM treatments. NCI/RES-ADR and A2780ADR cells were treated either with HM first (HM-DOX, cells were treated with HM 30 min earlier than DOX) or with both drugs together (HM+DOX, cells were treated with HM and DOX at the same time). As shown in the new Supplementary Fig. 27a and 27b for NCI/RES-ADR and A2780ADR cells, respectively, the fluorescence intensity of DOX in HM-DOX treated cells is similar to that in cells treated with both drugs simultaneously (HM+DOX) for 3h. Similarly, the cytotoxicity of the HM-DOX treatment is also not significantly different from that of the HM+DOX treatment (Supplementary Fig. 28a-b). These data suggest that releasing HM and DOX simultaneously from the FSCNO nanoparticles is of significance for overcoming cancer drug resistance. This is possibly because HM can quickly, selectively, and potently inhibit the P-gp function. As a result, although some DOX may be pumped out of cells before the function of all the P-gps is fully inhibited, it could re-enter the cells after the HM binds with all the P-gps quickly. The aforementioned discussions are now incorporated in last 7 lines in the third paragraph of Discussion section.

4. The Fucodan-based nanoparticle can target P-selectin for tumor accumulation. In addition, the P-selectin does not only appear on the tumor but also widely expressed on the vasculature in other organs. In this response, the authors do not reply the question when the FSCNO was subjected to Intravenous injection through the vessel, how the FSCNO can precisely delivered and targeted to tumor?

Re: We agree the FSCNO nanoparticles may bind with the P-selectin expressed in other normal tissues and not all the nanoparticles can accumulate at the tumor site. As shown in Figure 6c-d for the *in vivo* imaging, FSCNO nanoparticles could deliver more imaging agent to *in vivo* tumors if co-injecting the P-selection-overexpressed aHUVCEs with cancer cells for growing the tumors, suggesting the FSCNO nanoparticle can target P-selectin for enhanced tumor accumulation of the nanoparticles. Another mechanism is the enhanced permeability and retention (EPR) effect of tumor vasculature compared to the vasculature in normal tissues/organ, which has been widely used for developing nanomedicine to passively target tumor over normal tissues/organs. The aforementioned information is given in the last line on page 3 and first 5 lines on page 4. However, we understand the use of the word “precisely” for tumor targeting is too strong and change the title of the manuscript to “Carbon nano-onion-mediated dual targeting of P-selectin and P-glycoprotein to overcome cancer drug resistance” in this 2nd revision. The word “precisely” is no long used for tumor targeting throughout this manuscript. Again, we thank the reviewer for the insightful and thoughtful comments!

REVIEWERS' COMMENTS

Reviewer #1 (Remarks to the Author):

The authors have addressed all the questions adequately.

Reviewer #3 (Remarks to the Author):

The authors have tried to reply my concerns. Although the responses do not completely satisfy my questions, I think that it should be qualified to be published in this journal. Therefore, I do not have any further questions needing to be further replied.

Point-by-point response to reviewers

We would like to thank all the reviewers for their kind comments!

Reviewer #1

The authors have addressed all the questions adequately.

Re: We thank the reviewer for the kind comments!

Reviewer #3

The authors have tried to reply my concerns. Although the responses do not completely satisfy my questions, I think that it should be qualified to be published in this journal. Therefore, I do not have any further questions needing to be further replied.

Re: We thank the reviewer for the kind comments!